# Visualizing intraorganellar ultrastructures, dynamics, and interactions with open-access background-free Lock-in-SIM

Wenjie Liu [1,2,10] ✉, Meng Zhang [3,10], Wenbin Zhu [4,10], Shunyu Xie [5,10], Jinfeng Zhang [5,6], Shuhao Qian [5], Zhiyi Liu [5], Xiu Zheng [5,6], Qiuyuan Fang [7], Wei Yang [7], Yi Wang [8], Dazhao Zhu [4], Jianjie Dong [9], Xu Liu [5], Youhua Chen [5,6] ✉, Cuifang Kuang [5,6], Yu-Hui Zhang [3] ✉ & Lothar Schermelleh [1] ✉

Structured illumination microscopy (SIM) is a powerful method for fast and gentle live-cell super-resolution imaging. However, its susceptibility to reconstruction artifacts from out-of-focus blur and background imposes substantial barriers to analyze the dynamics of densely packed volumetric intraorganellar ultrastructures that are typically in a size range of SIM's spatial resolution. To address this limitation, we have developed Lock-in-SIM, an open-access two-dimensional SIM framework that eliminates background and maximizes the recovery of sub-diffraction information with the highest possible frequency extraction. By leveraging the intrinsic modulation differences of volumetric sample structures, Lock-in-SIM enables efficient optical sectioning, extends imaging depth, and improves data fidelity and quantifiability. We demonstrate the superiority of Lock-in-SIM by visualizing various challenging intraorganellar ultrastructures in live cells. Our investigations uncover mechanisms of mitochondrial fission and endoplasmic reticulum-lysosome interactions and provide insights into the intricate yet highly regulated structural remodeling of organelles.

Fluorescence super-resolution microscopy (SRM) has substantially enhanced our understanding of biological structures, dynamics, and functions by expanding the imaging capability of conventional optical microscopy from the diffraction-limited to subcellular, intraorganellar, and even molecular levels[1–4]. The visualization of ultrafine cellular components requires high spatial and temporal resolution, which has been the main direction of development in optical microscopy for decades. Structured illumination microscopy (SIM)[5–7] is particularly well-suited for live-cell super-resolution imaging, as it achieves a resolution doubling over conventional microscopy at a higher speed while imposing lower photodamage compared to other SRM approaches.

While SRM methods are typically compared by their maximum spatiotemporal resolution under ideal conditions, in real-world sample

[1]Department of Biochemistry, University of Oxford, Oxford, UK. [2]Kavli Institute for Nanoscience Discovery, University of Oxford, Oxford, UK. [3]MoE Key Laboratory for Biomedical Photonics, Advanced Biomedical Imaging Facility-Wuhan National Laboratory for Optoelectronics, Britton Chance Center for Biomedical Photonics, Huazhong University of Science and Technology, Wuhan, China. [4]Zhejiang Lab, Hangzhou, China. [5]State Key Laboratory of Extreme Photonics and Instrumentation, College of Optical Science and Engineering, Zhejiang University, Hangzhou, China. [6]Ningbo Research Institute, Zhejiang University, Ningbo, China. [7]NHC and CAMS Key Laboratory of Medical Neurobiology, Department of Biophysics, Institute of Neuroscience, Zhejiang University School of Medicine, Hangzhou, China. [8]Pharmaceutical Informatics Institute, College of Pharmaceutical Sciences, Zhejiang University, Hangzhou, China. [9]Zhangjiang Laboratory, Shanghai, China. [10]These authors contributed equally: Wenjie Liu, Meng Zhang, Wenbin Zhu, Shunyu Xie. ✉e-mail: wenjie.liu@chem.ox.ac.uk; chenyh21012@zju.edu.cn; zhangyh@mail.hust.edu.cn; lothar.schermelleh@bioch.ox.ac.uk

conditions, noise and background are major limiting factors that undermine the effective resolution, reduce image quality, and cause artifacts[8–11]. Although both can lead to the deterioration of the illumination light field and the overwhelming of weak high-frequency ultrastructures, the underlying mechanisms of influence are different[12,13]. Noise predominantly arises in the optoelectronic signal conversion processes of the detector, especially under the conditions of low exposure time or low illumination intensity, and commonly varies from pixel to pixel. By contrast, the background primarily originates from out-of-focus blur caused by epi-fluorescence illumination along the optical axis, particularly in densely packed volumetric samples, as well as light scattering, unspecific labeling, and auto-fluorescence. Background levels typically vary slowly across the field and are poorly modulated by the illumination pattern. While impressive advances have been made in suppressing noise in recent years[14–16], especially with the emergence of machine and deep learning technologies[17–22], more efforts are still required to specifically address the background issue in SRM. Furthermore, deep learning-assisted SRM methods commonly suffer from a lack of accessibility and universality due to their heavy reliance on high-quality training datasets[17]. This is, in part, why confocal microscopy, despite its lower acquisition speed, has become one of the gold standards for bioimaging applications, namely by effectively blocking background through a mechanical pinhole[8,23].

SIM's susceptibility to background artifacts is based on its reliance on accurate illumination patterning and the necessity for robust modulation, transmission, and demodulation of weak high-frequency signals to obtain high-quality results[24–27]. In addition, background from fluorescence labeling or autofluorescence might reduce the modulation contrast and interfere with the SIM imaging results[28,29]. Therefore, it remains challenging to investigate the ultrafast dynamics and interactions of volumetric and intraorganellar ultrastructures, such as the endoplasmic reticulum (ER) matrix[30] and mitochondria inner membrane[14], which play key roles in regulating biological health and disease. These ultrastructures have sizes that typically fall within the SIM resolution range and are thus prone to being obscured or misreconstructed by the presence of even slight background and motion[30]. Consequently, instead of the original two-beam interference two-dimensional structured illumination (2D-SIM) implementation[6], other SIM modalities using total internal reflection fluorescence (TIRF-SIM)[31], grazing incidence (GI-SIM)[30,32], and three-beam interference (3D-SIM)[33–35] have become more commonly used in practice due to their shared ability to effectively suppress background. However, the imaging depths of TIRF-SIM and GI-SIM are too limited to encompass entire cells, while 3D-SIM is typically subject to increased photobleaching and reduced temporal resolution due to the requirement of imaging z-stacks with 15 raw images (5 phases, 3 angles) per plane. Regarding 2D-SIM algorithms, substantial advances have been made in recent years to improve either reconstruction speed, image fidelity, or background suppression with methods such as weighted linear SIM (WLR-SIM)[36], Open-SIM[37], Fair-SIM[38], Hessian-SIM[14], inverse matrix SIM (IM-SIM)[39], shifting-phase SIM (SP-SIM)[40], high-fidelity SIM (HiFi-SIM)[15], joint space and frequency reconstruction SIM (JSFR-SIM)[41], principal component analysis SIM (PCA-SIM)[42], background-filtering SIM (BF-SIM)[43], Direct-SIM[44], and Flex-SIM[45]. However, current state-of-the-art 2D-SIM methods still face challenges in balancing background suppression and image fidelity (i.e., ultrastructure preservation and structural integrity), especially in densely packed intraorganellar imaging. Thus, there remains an unmet need for low-background, high-quality image data in many common imaging situations, which is a prerequisite for non-biased quantitative analysis[23,46,47].

To address these shortcomings, we propose a reconstruction framework for two-beam interference 2D-SIM, termed Lock-in-SIM, which achieves a signal-to-background ratio (SBR) comparable to that of TIRF-, GI-, and 3D-SIM but surpasses their limitations in temporal

resolution (3D-SIM) or imaging depth (TIRF-/GI-SIM). Lock-in-SIM exploits the modulation differences of the periodic SIM illumination pattern for sample structures of varying whole-cell depths to lock, extract, and amplify the weak high-frequency signal deriving from sub-diffraction-sized sample features while minimizing the effects of unmodulated background. This process is inherent in SIM imaging and can therefore be applied to data acquired from existing SIM systems, allowing for seamless integration into established post-processing workflows without additional hardware modifications and troublesome parameter fine-tuning.

The generalized implementation of Lock-in-SIM enables the background- and artifact-free visualization of a wide range of fixed and live subcellular and intraorganellar ultrastructures with the highest resolution, fidelity, and imaging depth theoretically attainable with SIM. Through quantitative long-term live-cell imaging, we demonstrate an improvement in quantification sensitivity and a reduction in analysis bias by utilizing Lock-in-SIM reconstruction. In two-color time-lapse ER-lysosome imaging, we successfully captured the structural rearrangements of intraorganellar components, including the ER matrix and lysosome tubules, and their interactions, shedding light on the remodeling and interactions of organelles. In multi-color time-lapse mitochondria-dynamin-related protein 1 (Drp1) imaging, we reveal a mechanism of mitochondria-mediated mitochondrial fission via the transfer of Drp1, completing the classical mitochondrial fission model and shedding light on the interactions of organelles. Lock-in-SIM combines compelling reconstruction quality with universal applicability, in a user-friendly open-access package, either as MATLAB GUI, Fiji/ImageJ plugin, or executable software, that will greatly facilitate the exploration of highly intricate and dynamic biological processes.

## Results
### Principle and validation of the background-free Lock-in-SIM
The key aspect of background removal is to identify and separate the signal from the background by detecting their differences. In electronic circuits, a classical method for separating signals and background noise involves the use of a lock-in amplifier. The amplifier generates a known modulated signal as the reference waveform that is highly correlated with the detecting signal but not with background noise. By performing correlation analysis between the reference and detecting waveforms, the amplifier effectively suppresses background noise and enables the extraction and tracking of weak signals. In this regard, upon reconsidering the imaging process of SIM, it becomes evident that the sinusoidal periodic illumination pattern of SIM can serve as an ideal reference signal. Based on this, the possibility of applying lock-in detection to widefield-based optical SIM imaging was explored.

The intensity distribution of a raw acquired SIM image $I_j(\mathbf{r})$ at a certain pattern orientation can be described by the following:

$$I_j(\mathbf{r}) = \left\{ \left[ 1 + m\cos\left(\mathbf{k}_0\mathbf{r} + \varphi_j\right) \right] \cdot O(\mathbf{r}) \right\} * H(\mathbf{r}) \qquad (1)$$

where $\mathbf{k}_0$ is the frequency coordinate, $\varphi_j$ is the phase of the $j$th phase-shifted illumination pattern ($j = 1,2,3$), $m$ is the modulation depth, $O(\mathbf{r})$ represents the sample information, $H(\mathbf{r})$ is the point spread function (PSF) of the system, and $*$ denotes the convolution operation.

In traditional 2D-SIM reconstruction, the sample $O(\mathbf{r})$ is regarded as a 2D structure that is modulated equidistantly at each lateral position by a sinusoidal illumination pattern. However, as the imaged sample and area are 3D distributed, the resulting raw image $I_j(\mathbf{r})$ is actually the intensity projection of the entire 3D volume, including both the in-focused sample structure and background. Figure 1a illustrates that the illumination pattern exhibits a relatively ideal sinusoidal distribution near the focal plane, which leads to better modulation of the in-focused sample structure. At the high-depth

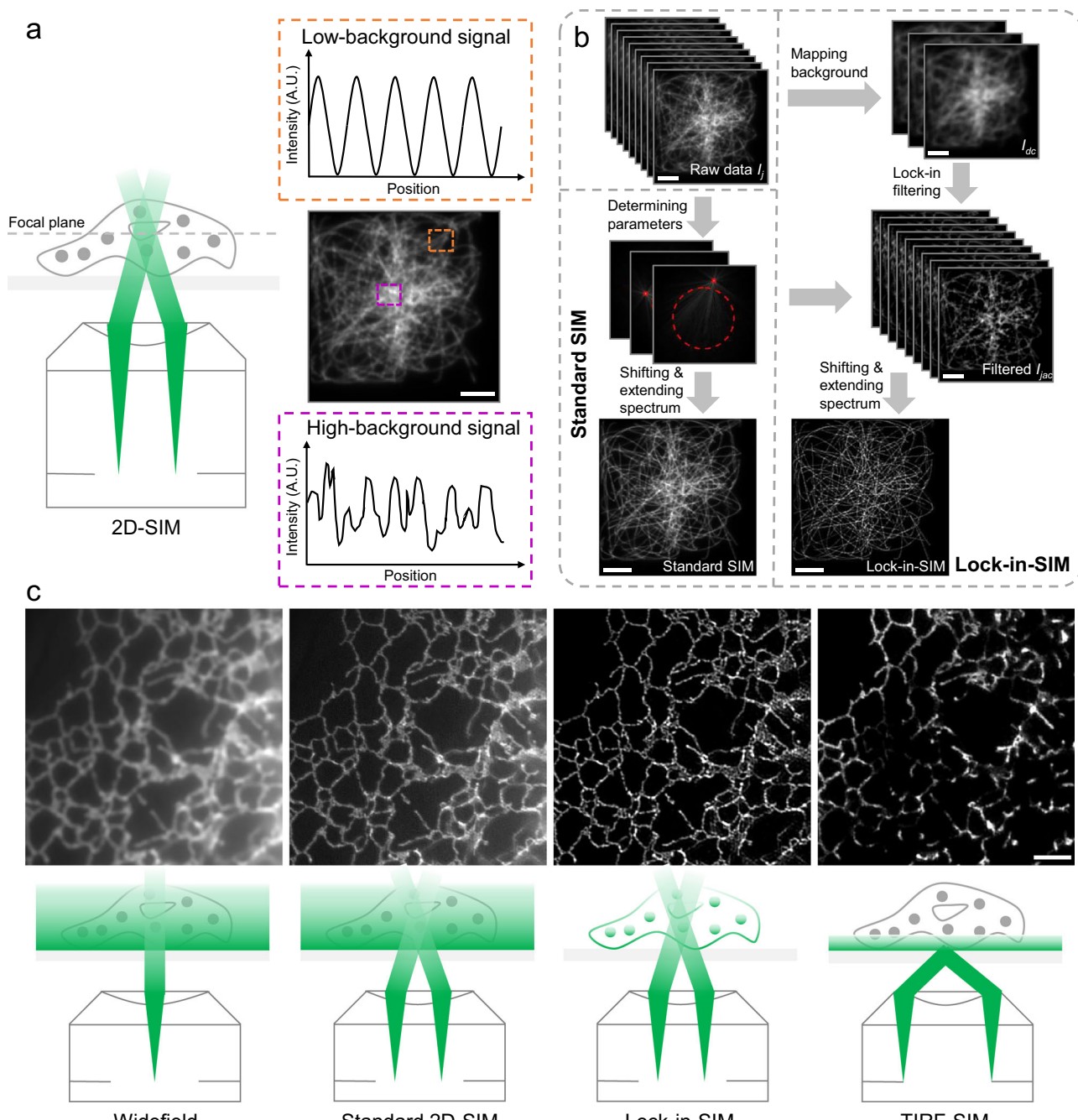

**Fig. 1 | Principle and validation of the background-free Lock-in-SIM. a** Left: schematic of the illumination beam positions at the back focal plane of the objective. Right: simulated widefield image at the focal plane (middle) and the schematic intensity distributions of the illumination pattern at the low-background (top, orange box) and high-background (bottom, purple box) positions. **b** Workflow of standard 2D-SIM and Lock-in-SIM reconstruction. The main difference is that the background $I_{dc}$ from the raw data $I_j$ is mapped and filtered out to obtain $I_{jdc}$ in Lock-in-SIM before the subsequent frequency shifting and extending procedures. **c** Experimental results (top) of Sec61β-EGFP-labeled ER using widefield

microscopy, standard 2D-SIM, Lock-in-SIM, and TIRF-SIM. The bottom schematics correspond to the illumination modes. The rectangular green region denotes the excitation area. In widefield and standard 2D-SIM, the fluorophores in the deep cell region are excited, which mainly contributes to the background. TIRF-SIM only collects the fluorescence signals within the shallow membrane area. Lock-in-SIM can retain the structural fluorescence signal within the entire 2D-SIM illumination area while eliminating background fluorescence. Images are representative of experiments conducted more than three independent times (**a**, **c**). Scale bar, 3 μm (**a**, **b**) and 2 μm (**c**).

positions, the modulation effect of the illumination stripe rapidly deteriorates due to the increasing poorly modulated and unmodulated background from scattering, autofluorescence, noise, and aberration. Therefore, Eq. (1) should be rewritten as follows:

$$I_j(\mathbf{r}) = \left\{ O_{out}(\mathbf{r}) + \left[ 1 + m\,cos\left( \mathbf{k}_0 \mathbf{r} + \varphi_j \right) \right] \cdot O_{in}(\mathbf{r}) \right\} * H(\mathbf{r}) \quad (2)$$

considering the different contributions of the unmodulated background $O_{out}(\mathbf{r})$ and in-focused signal $O_{in}(\mathbf{r})$. Then, cross-correlation[48] and frequency analysis[49] could be considered to separate $O_{out}(\mathbf{r})$ and $O_{in}(\mathbf{r})$. However, these analysis pipelines are time-consuming and not suitable for SIM applications because multiple modulation cycles with sufficient sampling steps are needed to acquire tens to hundreds of raw images for precise separation (Supplementary Fig. 1 and Note 2).

To address this issue, here, we further rewrite Eq. (2) as the sum of two components, the modulation-independent (DC) background $I_{dc}(\mathbf{r})$ and modulation-dependent (AC) signal $I_{j_{ac}}(\mathbf{r})$:

$$I_j(\mathbf{r}) = \underbrace{\left[O_{out}(\mathbf{r}) + (1-m)\cdot O_{in}(\mathbf{r})\right] * H(\mathbf{r})}_{I_{dc}(\mathbf{r})}$$
$$+ \underbrace{\left[m + m\cos\left(\mathbf{k}_0\mathbf{r} + \varphi_j\right)\right]\cdot O_{in}(\mathbf{r}) * H(\mathbf{r})}_{I_{j_{ac}}(\mathbf{r})} \quad (3)$$

The DC component $I_{dc}(\mathbf{r})$ can be calculated with raw images $I_j(\mathbf{r})$ at three phases (see Supplementary Note 1 for further algorithm details). The in-focused signal is then obtained by filtering out $I_{dc}(\mathbf{r})$ from the raw image $I_j(\mathbf{r})$, which is subsequently used for super-resolution SIM reconstruction (Fig. 1b).

Figure 1c shows the experimental results from the different 2D imaging methods. Although the subtle hollow structures of the ER were distinguishable after standard 2D-SIM reconstruction, they were superimposed with high background, and thus, the image contrast, i.e., effective resolution, deteriorated. In contrast, TIRF-SIM was capable of providing background-free and high-contrast results due to the subwavelength-depth illumination, but the ER structures beyond the illumination depth were lost. With our Lock-in-SIM, the background-depth trade-off was broken. Lock-in-SIM maintained both TIRF-level contrast and 2D-level structural integrity and therefore could facilitate new advances for cell biology research. To further illustrate the capability and biological benefits of Lock-in-SIM, we focused on the ER in thicker perinuclear regions of the cell (Supplementary Fig. 2) rather than examining only the cell margin. By merging Lock-in-SIM with standard 2D-SIM (Supplementary Fig. 2e) and TIRF-SIM (Supplementary Fig. 2f), Lock-in-SIM clearly preserved the entire ER meshwork while eliminating the background, especially around thicker, more densely packed perinuclear ER.

## Lock-in-SIM provides optimal background suppression, image resolution, and fidelity

To systematically test the performance of the algorithm, we further compared Lock-in-SIM with a total of 13 existing algorithms, including Wiener-SIM[6], WLR-SIM[36], Open-SIM[37], Fair-SIM[38], Hessian-SIM[14], IM-SIM[39], SP-SIM[40], HiFi-SIM[15], JSFR-SIM[41], PCA-SIM[42], BF-SIM[43], Direct-SIM[44], and Flex-SIM[45]. Applied to simulated, synthetic, or experimental bead datasets, Lock-in-SIM reconstructions proved highly robust against variations in noise (Supplementary Fig. 3), field-of-view distortion (Supplementary Fig. 4), and background (Supplementary Figs. 5 and 6), while the algorithms compared against showed different degrees of susceptibility to these factors. Notably, the resolution of Lock-in-SIM results remained consistently high across varying background conditions, closely matching the theoretical SIM resolution (Supplementary Fig. 6). When applied to 3D volumetric datasets, Lock-in-SIM effectively eliminated out-of-focus background to provide optical sectioning and deliver high-quality pseudo-3D reconstructions from standard nine-image 2D-SIM acquisition (Supplementary Fig. 3 and Supplementary Movie 1).

We next validated the superiority of Lock-in-SIM for different sample types with zero-, one-, two-, or three-dimensional continuity and varying imaging conditions using both open-access (Supplementary Figs. 7–13) and our own experimental datasets (Fig.2, Supplementary Figs. 14–20, and Supplementary Movie 2), including the mitochondrial inner and outer membranes, microtubules, actin, lysosomes, clathrin-coated pits, mitochondrial-derived vesicles, and 5-ethynyl-2′-deoxyuridine (EdU) replication-labeled nuclei. In all these scenarios, Lock-in-SIM outperformed the other algorithms by effectively removing background and retaining the in-focus ultrastructure with high integrity, achieving superior image definition and reconstruction fidelity. The global SBR (Fig. 2e, g and Supplementary

Figs. 14b and 15b) and local SBR maps (Supplementary Fig. 14a) quantitatively supported the above deductions. As the background was suppressed, the effective resolution was improved (Fig. 2f, h). Accordingly, the membranes of mitochondria (Fig. 2a and Supplementary Figs. 7, 15 and 17) and the hollow structures of clathrin-coated pits (Supplementary Fig. 18) were more clearly visible after Lock-in-SIM reconstruction.

The background fluorescence may also lead to reconstruction artifacts in addition to blurring the cell structure of interest[25,26]. As shown in Fig. 2c, d, evident ringing and hexagonal (honeycomb) artifacts appeared around the structure in the existing 2D-SIM images, while these artifacts could be suppressed through the Lock-in-SIM reconstruction. When reconstructing high-background and high-density samples, existing methods often compromise between SBR and structural integrity (Supplementary Figs. 14 and 16). In contrast, Lock-in-SIM achieved optimal background suppression and structural fidelity. When inspecting the frequency distribution (Supplementary Fig. 14), the reconstructed spectra of existing 2D-SIM showed abnormal amplitude drop-offs in both the low-frequency and high-frequency bands; these correspond to the possible image artifacts in the spatial domain. In contrast, the spectrum of the Lock-in-SIM image decayed more smoothly. Lock-in-SIM had the smallest slope and the highest amplitude and cut-off frequency, which indicated higher signal strength, noise resistance, and effective resolution. Therefore, considering either the space or frequency domain under diverse imaging conditions, our Lock-in-SIM extracts weak high-frequency signals, preserves ultrastructural integrity, and extends the frequency transfer function with the highest fidelity (Supplementary Note 3).

## High-fidelity whole-cell optical sectioning reconstruction using Lock-in-SIM with standard 2D-SIM acquisition

In general, 2D SRM techniques, including 2D-SIM, improve only the lateral resolution and lack optical sectioning ability due to the influence of the background. Although 3D-SIM can solve this problem, 15 raw images at each focal plane (3 angles and 5 phases), and at least ±3 adjacent planes are required for reconstructing each single plane, which results in a lower acquisition speed and stronger photobleaching[35,46,50]. Optical sectioning-SIM suppresses the background at the cost of lower spatial resolution[24]. Here, we show that Lock-in-SIM can alleviate the imaging trade-off among optical sectioning, spatial resolution, and temporal resolution.

In addition to the simulated 3D volumetric filament structure (Supplementary Fig. 3, Supplementary Movie 1), we performed axial scanning experiments on single-color microtubules, mitochondria, EdU-labeled replication sites, and two-color microtubule-lysosomes with normal nine-image 2D-SIM acquisition at each axial plane to demonstrate the optical sectioning capability of the Lock-in-SIM (Fig. 3, Supplementary Figs. 20 and 21, and Supplementary Movies 3–7). In all cases, Lock-in-SIM resulted in much greater SBR than the standard 2D Wiener-SIM reconstruction. By color-coding the pixel intensities of each stack, Lock-in-SIM provided both super-resolution and optical-sectioning depth information, whereas this information was highly obscured by the background in the standard 2D-SIM results (Fig. 3 and Supplementary Movies 3 and 4). When comparing the whole-cell-depth microtubule datasets acquired at different stages of growth (Fig. 3a–d), the cells at the telophase (Fig. 3c) extend over a greater depth range, approximately twice that of the non-telophase cell (Fig. 3a). Interestingly, by applying orientation analysis[51,52] (Fig. 3n; see the next section and "Methods" for details), we found that the non-telophase microtubules tended to be uniformly aligned towards the cell migration direction, while the telophase cell exhibited a two-peak orientation distribution spaced nearly 90° apart (i.e., most microtubules were orthogonally organized); these results indicated the differences in microtubule function at different growth stages. The high-depth axial view of the telophase

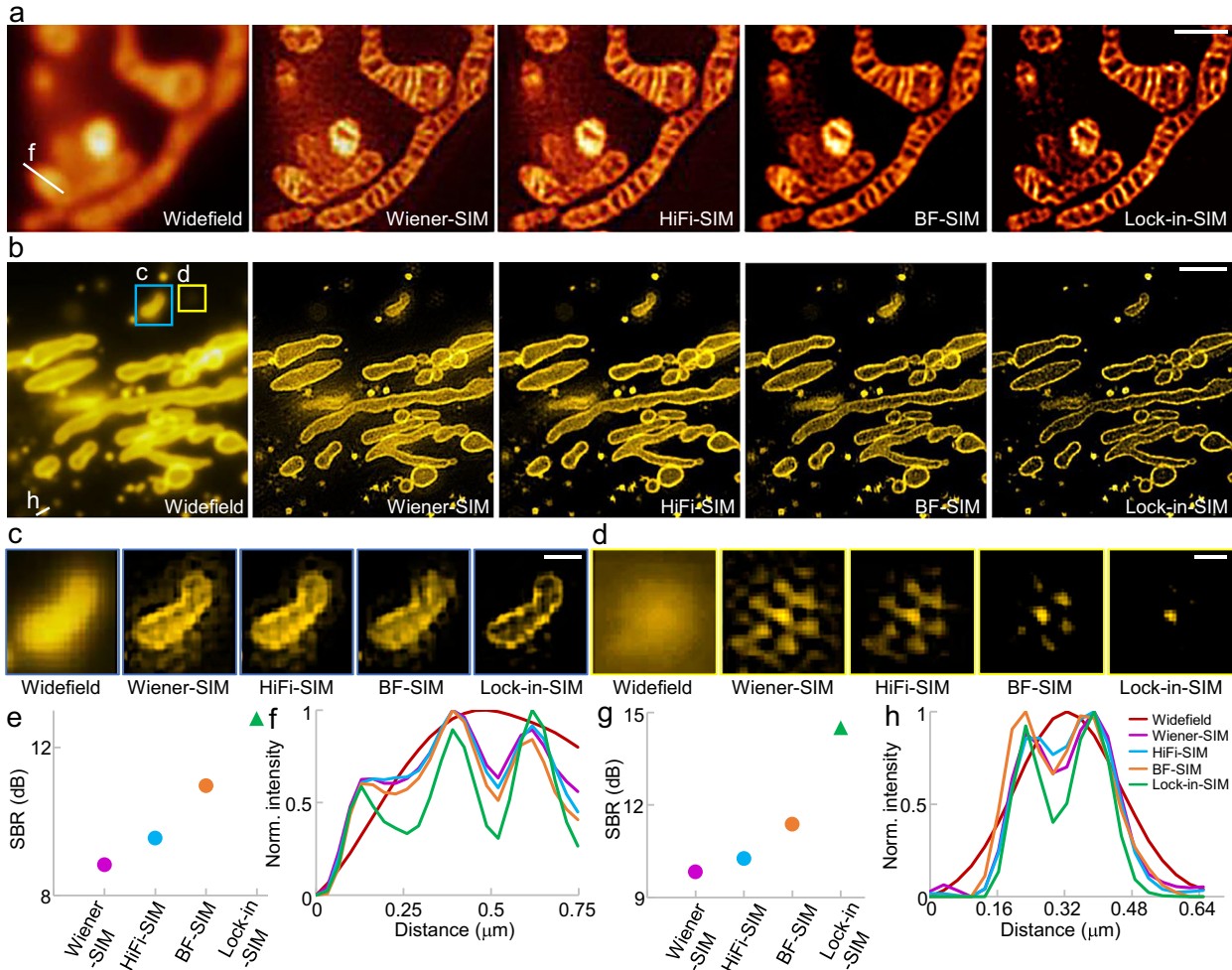

**Fig. 2 | Lock-in-SIM provides optimal background suppression, image resolution, and fidelity. a** Widefield image of the MitoTracker Green FM-labeled mitochondrial inner membrane and super-resolution images reconstructed using Wiener-SIM, HiFi-SIM, BF-SIM, and Lock-in-SIM. **b** Widefield image of the Tomm20-EGFP-labeled outer mitochondrial membrane and super-resolution images reconstructed using Wiener-SIM, HiFi-SIM, BF-SIM, and Lock-in-SIM. Magnified images of the blue (**c**) and yellow (**d**) boxed regions in (**b**), showing the background and artifact suppression with Lock-in-SIM reconstruction. **e** SBR values of the super-resolution images reconstructed by the different algorithms in (**a**). **f** Intensity profiles along the white lines in (**a**). Profile legends are the same as those of (**h**). **g** SBR values of the super-resolution images reconstructed by the different algorithms in (**b**). **h** Intensity profiles along the white lines in (**b**). Images are representative of experiments conducted more than three independent times (**a–c**). Scale bars, 1 μm (**a**), 2 μm (**b**), 0.5 μm (**c**), and 0.3 μm (**d**).

microtubules further strongly confirmed the improvement in the optical sectioning capability of standard 2D-SIM by using Lock-in reconstruction (Fig. 3d). The reconstruction comparison of different mitochondrial depths shown in Fig. 3g and Supplementary Movies 6 and 7 demonstrates Lock-in-SIM's capability of efficiently filtering out the high background and retaining the depth-dependent structural features at each axial plane, while the structure and background were mixed in Wiener-SIM images. According to the two-color microtubule-lysosome data (Fig. 3h–j and Supplementary Movie 4), Lock-in-SIM also provided background-free depth information and super-resolution structural details to facilitate colocalization studies, while standard 2D-SIM only improved the resolution of the widefield images.

Intensity profile plots along adjacent fluorescent features not resolvable by widefield also display far higher peak-to-valley intensity values with Lock-in-SIM, further validating contrast and effective resolution enhancement (Fig. 3k–m). Benchmarking Lock-in-SIM against standard 3D-SIM of the same cell (Supplementary Figs. 21 and 22 and Supplementary Movie 8), both displayed highly consistent depth and 3D structural distributions, thus confirming the reconstruction effectiveness and accuracy of our Lock-in-SIM method.

In summary, our Lock-in-SIM provides a powerful tool for achieving 3D-SIM-like high-fidelity whole-cell volumetric super-resolution imaging with standard 2D-SIM acquisitions. Compared to 3D-SIM, Lock-in-SIM has similar optical sectioning capability, i.e., similar SBR and structural depth distribution, albeit without two-fold axial resolution improvement (Supplementary Figs. 21 and 22), while improving the imaging speed beyond 3-fold due to its requirement for 9 versus 15 exposures and halved z-sampling, which greatly reduces photobleaching and phototoxicity. In addition, Lock-in-SIM's ability to achieve optical sectioning from a single-plane acquisition renders it ideal for ultrafast, background-free 2D time-lapse live-cell measurements, which are not possible with either standard 2D-SIM or 3D-SIM.

## Non-biased quantitative live-cell imaging and analysis using Lock-in-SIM

One of SIM's most important applications is live-cell super-resolution imaging, which benefits from its gentle illumination intensity and high acquisition speed compared to other SRM approaches. However, the background is more prone to interfere with live-cell SIM imaging results. To overcome the effect of a high level of unmodulated bright background, one may need to increase the dynamic range by

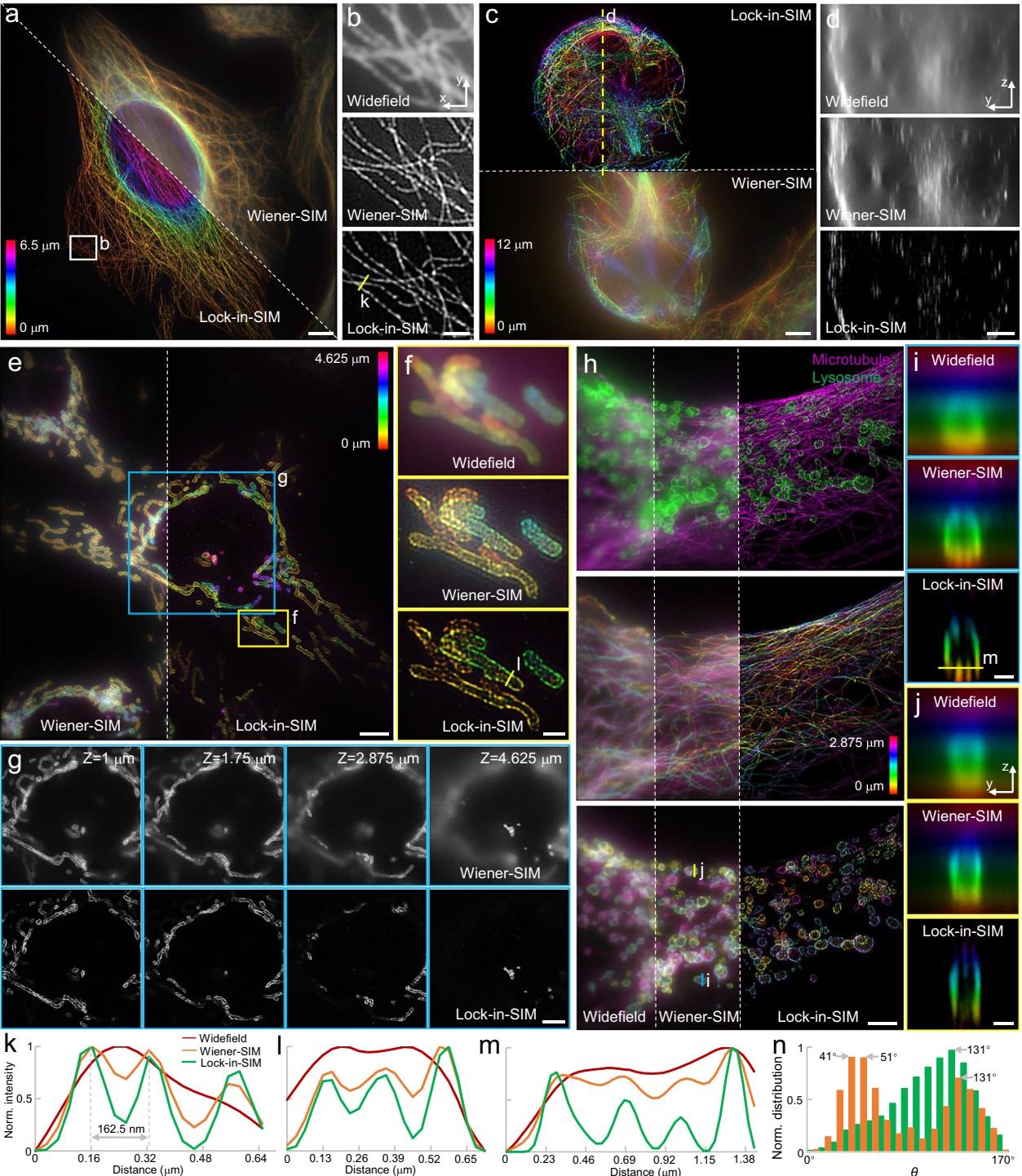

**Fig. 3 | High-fidelity whole-cell optical sectioning reconstruction using Lock-in-SIM with standard 2D-SIM acquisition. a** Depth-coded maximum intensity projection of a 6.5-μm thick 2D-SIM image stack of an anti-tubulin-labeled U2OS cell using Wiener-SIM (top right) and Lock-in-SIM (bottom left) reconstruction.
**b** Magnified images of the white-boxed region in (**a**). **c** Depth-coded maximum intensity projection of a 12-μm thick image stack of a telophase cell reconstructed using Wiener-SIM (bottom) and Lock-in-SIM (top). The cell is at the telophase.
**d** Axial views along the yellow dashed line in (**c**). **e** Depth-coded maximum intensity projection of a 4.625-μm thick image stack of anti-Tomm20-labeled mitochondria reconstructed using Wiener-SIM (left) and Lock-in-SIM (middle). **f** Magnified images of the yellow boxed region in (**e**). **g** Reconstructed results of different depths of

the blue boxed region in (**e**). **h** Two-color (top) and depth-coded maximum intensity projection of a 2.875-μm thick image stack antibody-labeled microtubules (middle) and anti-rab7-labeled lysosomes (bottom) in widefield (left), and after using Wiener-SIM (middle) and Lock-in-SIM (right) reconstruction. Axial views along the blue (**i**) and orange (**j**) dashed lines in (**h**). **k**–**m** Normalized intensity profiles along the yellow lines in (**b**, **f**, **i**), respectively. The profile legends of (**l**, **m**) are the same as those of (**k**). **n** Orientation distribution of the cell is shown in (**a**) (green histogram) and in (**c**) (orange histogram). Images are representative of experiments conducted more than three independent times (**a**–**j**). Scale bars, 5 μm (**a**, **e**), 1 μm (**b**, **f**), 3 μm (**c**, **d**, **h**), 4 μm (**g**), and 0.5 μm (**i**, **j**).

increasing laser power and exposure time, albeit at the cost of higher photobleaching. In addition, quantitative biological analysis, which is an increasingly important aspect in elucidating structure-dynamic-function relationships, is also susceptible to background noise[23,46,47]. Here we show that, benefiting from more efficient background subtraction, Lock-in-SIM can extend sample viability and provide higher-quality reconstruction results for quantitative live-cell imaging and analysis.

We first tested Lock-in-SIM with Tubulin-488-labeled living U2OS cells by recording long-term microtubule dynamics over 7 min (Fig. 4a–f, Supplementary Figs. 23 and 24 and Movie 9). Compared to Wiener-SIM, Lock-in-SIM eliminated the background effectively and thus facilitated detailed dynamic observation. We then mapped the orientation distribution ($\theta$ angle in Fig. 4b), which is a valuable quantification metric for reporting the cell polarity[51,52] (see "Methods" for algorithm details), of the microtubules to determine the spatial organization of the cell. As shown in Fig. 4b, the microtubule orientation, especially around the dense cell region, was difficult to extract based on the standard 2D-SIM result due to the influence of the background, while Lock-in-SIM reconstruction was able to generate unambiguous, high-precision $\theta$ mapping. Figure 4c–e shows the time-lapse intensity and orientation images of representative microtubules; these results demonstrate the reconstruction robustness of Lock-in-SIM at different structural density levels. In contrast, the standard 2D-SIM image could lead to inaccurate analysis results due to the influence of the background (Supplementary Fig. 23). By plotting the evolution curve of the $\theta$ angle, we found that the cell presented different orientation values and remodeling behaviors in different cell regions in response to the corresponding cellular functions (Fig. 4f). Specifically, the orientation of the dense cell region (green curve in Fig. 4f) varied more significantly than that of the sparser regions. In addition, the $\theta$ value gradually decreased in some regions (orange curves in Fig. 4f), while in others, it fluctuated around a certain baseline (blue curve in Fig. 4f).

When observing living mitochondria (Fig. 4g–l), Lock-in-SIM again achieved high-fidelity reconstruction with more clearly discriminated inner membrane structures. Based on the Lock-in-SIM images, the orientation evolution of mitochondria could be visualized and analyzed at the single cristae level. However, this information was mixed with the background in standard 2D-SIM results (Fig. 4h), resulting in quantification errors (Fig. 4k). Tracking a single mitochondrion, despite deforming distinctly in shape, its global orientation remained stable (Fig. 4l). In summary, our Lock-in-SIM reaches the full SBR, fidelity, sensitivity, and accuracy potential of both image reconstruction and quantification algorithms and thus greatly promotes the application of SIM towards more quantitative and dynamic live-cell studies.

## Visualization of the intraorganellar ultrastructures, dynamics, and interactions using Lock-in-SIM

After establishing the effectiveness of Lock-in-SIM in live-cell imaging, we moved on to more challenging biological imaging tasks. The densely and minutely packed matrices of the ER have long been misinterpreted as flat sheets by standard 2D-SIM due to insufficient spatial and temporal resolution, as well as 3D-projected background blurring[30]. With Lock-in demodulation, we showed that the ER matrix structure and dynamics can be clearly visualized in a 2D context in the presence of high background (Supplementary Figs. 25 and 26, and Supplementary Movies 10 and 11). Thanks to the highly effective spatiotemporal resolution of Lock-in-SIM, we were able to perform long-term tracking of the intricate, drastic, and rapid morphological distortion of both the outer and inner structures of the lysosomes. During movement, the lysosome could either maintain a rounded shape (Supplementary Movie 11) or evolve in a wide variety of forms, such as fusiform, drop-shaped, wheel-like, double-ring, figure-eight, and furcate (Supplementary Fig. 27), to timely adapt to changes in cell function. Additionally, two-color time-lapse Lock-in-SIM images of

lysosomes and ER showed that the lysosomes could regulate not only ER tubule[53] but also the dense ER matrix and that lysosomes moved at a lower speed than the commonly reported movements in non-matrix tubular regions (Supplementary Fig. 28 and Supplementary Movie 11).

The tubules of the lysosomes (tubular structures that grow from the lysosome), which play crucial roles in lysosomal trafficking and vesicle sorting[54,55] (Supplementary Figs. 25, 29 and 31 and Movies 10 and 11), are characterized by their relatively fine diameter and high dynamics; thus, they are also challenging to image with conventional imaging techniques. By clearly imaging the lysosome tubule, two-color time-lapse Lock-in-SIM validated the phenomenon of ER-mediated lysosome tubule fission[54] and further revealed that tubules were detached from whole lysosomes at different contact sites by their interactions with the ER tubules (Supplementary Figs. 25, 30 and 31, and Movie 11). Quantitative analysis indicated that the lysosome tubule fission tended to occur at the root and middle locations for the short (mean length = 0.65 μm) and long tubules (mean length = 7.85 μm), respectively (Supplementary Fig. 25e). After fission, the detached tubules moved ~4 times faster (mean velocity = 0.81 μm/s) than both the other part and the original lysosomes (Supplementary Fig. 25f).

Lock-in-SIM finally helps unveil insights into the dynamic behavior of mitochondria. Time-lapse imaging of the inner mitochondrial membrane captured the dynamics of mitochondrial cristae structures, such as mitochondrial fusion and fission, as well as inner membrane fusion and fission, with unprecedented image definition (Fig. 5, Supplementary Fig. 32, and Supplementary Movie 12). Interestingly, when observing two independent mitochondria in close proximity for an extended period, the super-resolution Lock-in-SIM images showed the existence of intricate membrane connections between them (Fig. 5b, Supplementary Fig. 33, and Supplementary Movie 13). This observation indicated that mitochondria potentially exchanged materials across these fine membrane junctions.

Mitochondrial fission is a vital bioprocess closely linked to proper cellular functioning[56,57] and has been previously described as being regulated only by other types of organelles or cytoskeletons, mainly the ER[32,58] and actin[59]. However, based on the long-term dynamic Lock-in-SIM imaging of PKmito DEEP RED-labeled living mitochondria, we discovered mitochondria-mediated mitochondrial fission (MMF), a mechanism distinct from the classical mitochondrial fission model. In the MMF mechanism, mitochondrial fission occurred via interactions between the mitochondria themselves (Fig. 5e, f, Supplementary Figs. 34 and 35 and Movie 13) rather than through interactions between different subcellular components (Fig. 5d). To further investigate the MMF mechanism and understand the underlying bioprocesses, we performed dual-color live-cell Lock-in-SIM imaging of the Tomm20-EFGP-labeled mitochondrial outer membrane and mCherry-labeled Drp1 oligomers. The results showed that during the MMF process, one mitochondrion first contacted another mitochondrion and initiated its fission. Thereafter, two Drp1 oligomers, located separately on the outer membrane of the two mitochondria, moved towards the contact site simultaneously and then merged to drive the final scission (Fig. 5g, Supplementary Fig. 36, and Supplementary Movie 14). These results improved the prevailing mitochondrial fission model by uncovering the MMF mechanism consisting of several kinetically distinct steps: the contact of two mitochondria, the transfer of Drp1 from the non-dividing mitochondrion to the dividing mitochondrion (along with the contact site-directed movement of Drp1 located on the dividing mitochondrion), and the fusion of the Drp1 oligomers to produce eventual mitochondrial division (Fig. 5h). Lock-in-SIM tracked the entire self-interaction processes of the mitochondria at high spatio-temporal resolution and elucidated the key role of Drp1[60] in MMF. To our knowledge, this is the first report to demonstrate that mitochondrial fission can also be regulated by the mitochondria themselves, providing insights into mitochondrial and cellular function. Taken

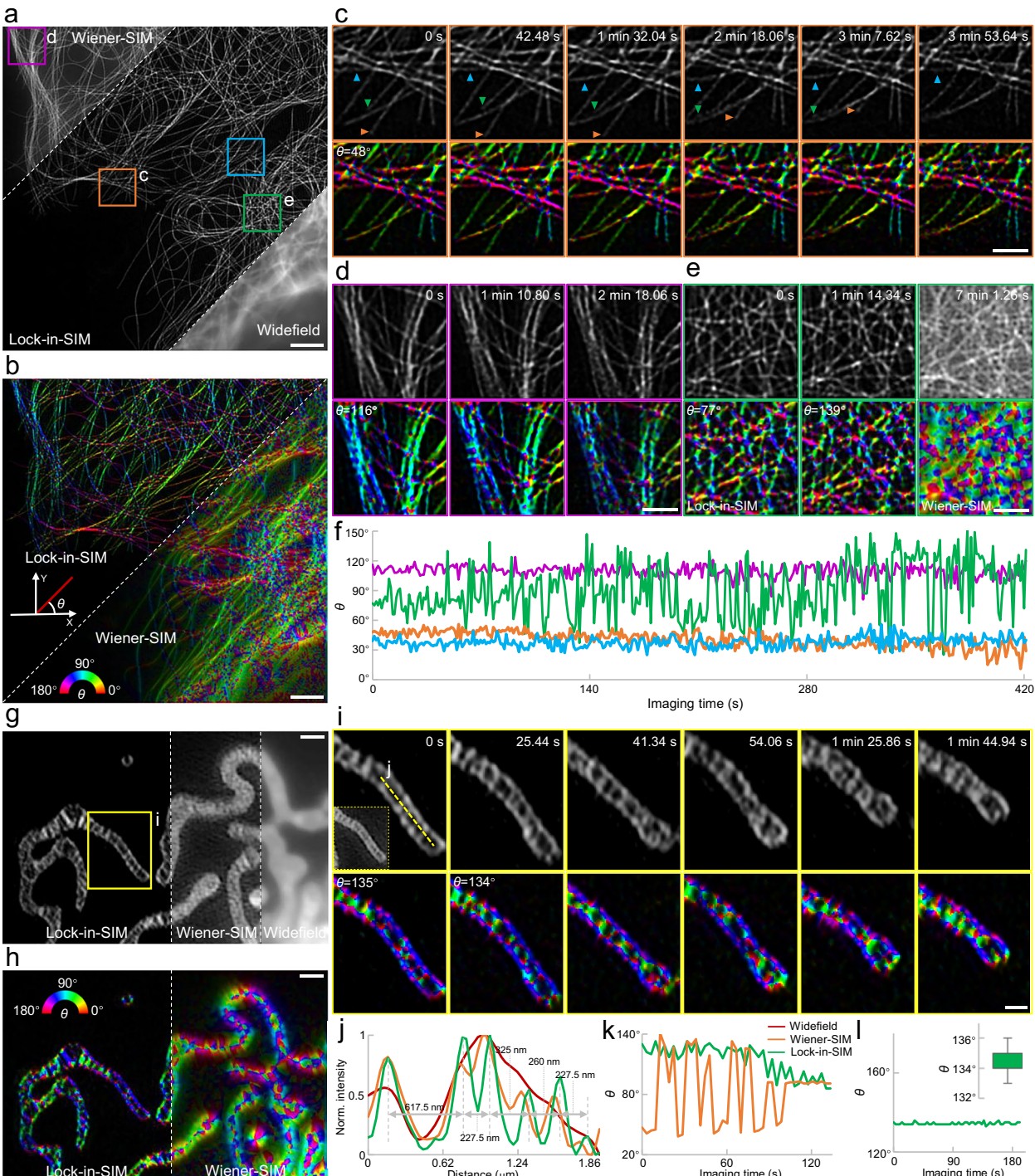

**Fig. 4 | Non-biased quantitative live-cell imaging and analysis using Lock-in-SIM. a** Widefield (bottom right) and super-resolution images reconstructed using Wiener-SIM (top left) and Lock-in-SIM (middle left) of Tubulin-488-labeled microtubules. **b** Orientation distribution of the reconstructed Lock-in-SIM (top left) and Wiener-SIM (bottom right) images in (**a**). The angle between the tubule and the coordinate system, $\theta$, is calculated and color-coded to denote its spatial orientation. Magnified time-lapse Lock-in-SIM images and their corresponding orientation distributions of the orange (**c**) and purple (**d**) boxed regions in (**a**). The tagged $\theta$ value is the $\theta$ with the highest distribution probability. Arrowheads in (**c**) indicate microtubules swinging (blue), growing (green), and contracting (orange). **e** Magnified time-lapse Lock-in-SIM (left and middle) and Wiener-SIM (right) images and their corresponding orientation distributions of the green boxed regions in (**a**). **f** Time-lapse Lock-in-SIM $\theta$ fluctuation of the corresponding color boxed regions in (**a**). **g** Widefield image (right) of PKmito DEEO RED-labeled living mitochondria and

super-resolution images reconstructed using Wiener-SIM (middle) and Lock-in-SIM (left). **h** Orientation distribution of the reconstructed Lock-in-SIM (left) and Wiener-SIM (right) images in (**g**). **i** Magnified time-lapse Lock-in-SIM images and their corresponding orientation distributions of the yellow boxed regions in (**g**). The inset is the corresponding Wiener-SIM result at $t = 0$ s. **j** Normalized intensity profiles along the yellow dashed line in **i**. Profile legends are the same as those of (**k**). **k** Time-lapse Lock-in-SIM (green) and Wiener-SIM (orange) $\theta$ fluctuation of the images in (**g**). **l** Time-lapse Lock-in-SIM $\theta$ fluctuation of the yellow boxed region in (**g**). The inset is the corresponding boxplot ($n = 61$ time points). The upper and lower boundary lines correspond to the 15th and 75th percentiles, respectively. The whiskers represent the min and max values. Images are representative of experiments conducted more than three independent times (**a**, **b**, **g**, **h**). Scale bars, 3 μm (**a**, **b**), 1 μm (**c**–**e**, **g**, **h**), and 0.5 μm (**i**).

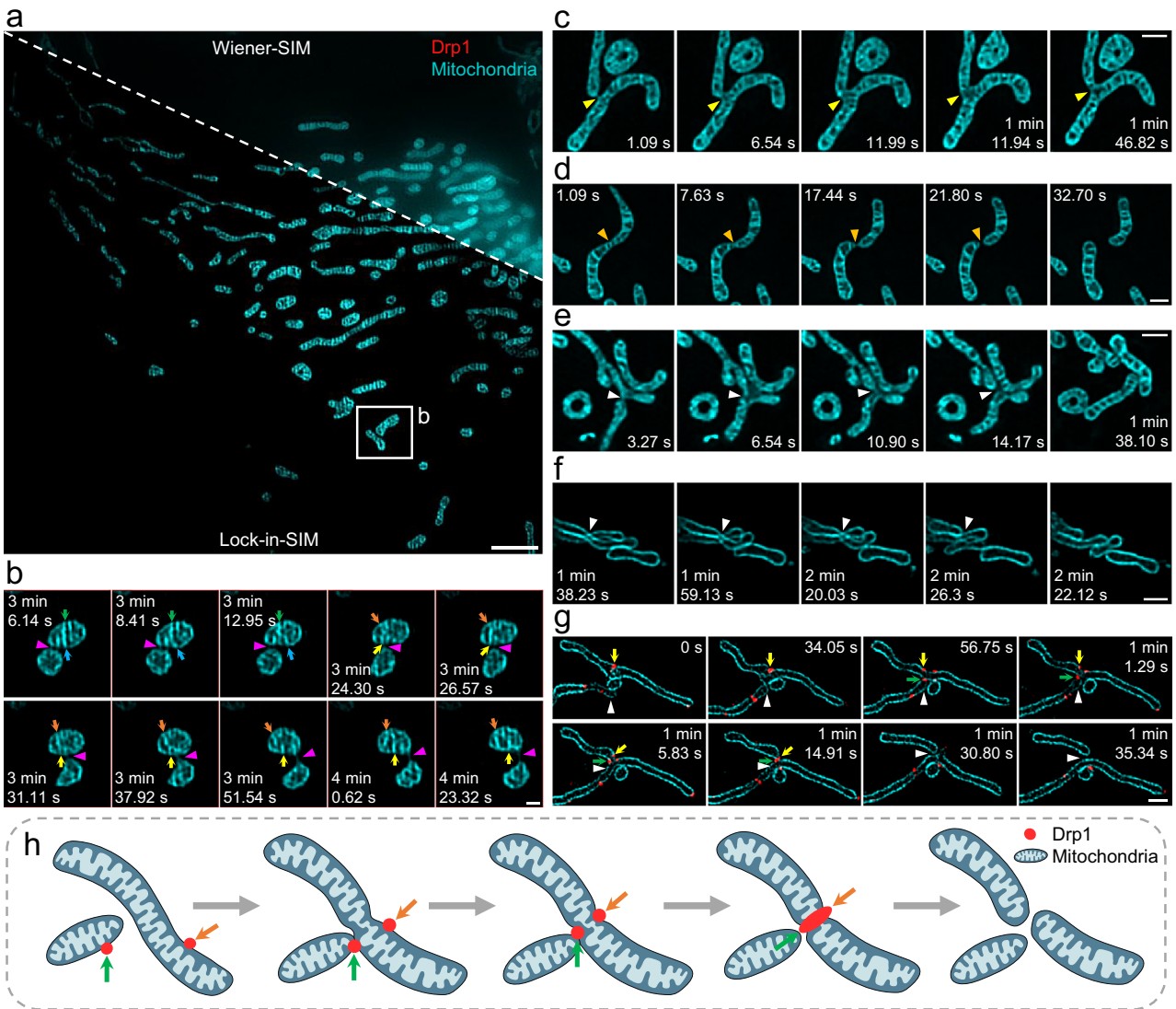

**Fig. 5 | Live-cell imaging of mitochondrial ultrastructures, dynamics, and interactions using Lock-in-SIM. a** Wiener-SIM (top right) and Lock-in-SIM (bottom left) images of PKmito DEEP RED-labeled mitochondria inner membranes. **b** Time-lapse Lock-in-SIM images of the white-boxed region in (**a**). Purple arrowheads indicate mitochondrial movement along another mitochondrion. The pairs of green and blue arrows and orange and yellow arrows indicate the relative movement between mitochondrial cristae structures, including fission and fusion. **c** Time-lapse Lock-in-SIM images of mitochondrial fusion. Yellow arrowheads indicate the fusion site. **d** Time-lapse Lock-in-SIM images of mitochondrial fission without another mitochondrial mediation. Orange arrowheads indicate the fission site. **e, f** Time-lapse Lock-in-SIM images of mitochondrial fissions with another mitochondrial mediation (MMF). The imaging samples were PKmito DEEP RED-labeled mitochondria inner membranes (**e**) and Tomm20-EFGP-labeled mitochondrial outer membranes (**f**). White arrowheads indicate the MMF sites. **g** Two-color time-lapse Lock-in-SIM images of Tomm20-EFGP-labeled mitochondrial outer membrane (cyan) and mCherry-labeled Drp1 oligomers (red). White arrowheads indicate the MMF event. Yellow arrows indicate the Drp1 oligomer on the dividing mitochondria. Green arrows indicate the Drp1 oligomer transported by the other mitochondria to participate in the MMF. **h** Schematic showing the MMF mechanism via the transport of Drp1. Orange arrows indicate the Drp1 oligomer on the dividing mitochondria. Green arrows indicate the Drp1 oligomer transported by the other mitochondria to participate in the MMF. Images are representative of experiments conducted more than three independent times (**a–g**). Scale bars, 3 µm (**a**), 0.5 µm (**b**), and 1 µm (**c–g**).

together, the presented results demonstrate the usefulness of Lock-in-SIM for advanced cell biology investigations.

## Discussion

We developed a user-friendly, open-access Lock-in-SIM framework to achieve generally opposing goals in super-resolution live-cell imaging: high-fidelity, highly quantifiable, extended duration, high spatio-temporal resolution, whole-cell, and 3D optical sectioning with effective background suppression. Compared to TIRF-SIM and GI-SIM, Lock-in-SIM provides ~100× and ~10× greater imaging depths, respectively, allowing for easy capturing of entire cell volumes. In comparison to existing 2D-SIM reconstruction algorithms, Lock-in-SIM simultaneously features optimal background suppression and image fidelity

(i.e., structural integrity, ultrastructure preservation, and definition) without any trade-offs, whereas, compared to 3D-SIM, Lock-in-SIM captures images >3× faster and allows single z optical sectioning (Supplementary Table 1). Importantly, these advances are achieved without imposing either additional imaging and post-processing burdens or trade-offs in any other 2D-SIM metrics, such as system/sample universality, photobleaching/phototoxicity, and multi-color imaging. These metrics are equally important for biological imaging but are normally compromised in state-of-the-art SRM[3,61]. The combined properties render Lock-in-SIM a timely complement of the prevailing deep learning-based SIM route and an ideal method for investigating volumetric intracellular and intraorganellar structures and dynamics. In addition to the remarkable scientific advancements, the

commercialization of SIM is also thriving. As an open-access algorithm, Lock-in-SIM can fully support the broader range of commercial linear SIM users. We additionally provide a Fiji/ImageJ plugin for converting various raw SIM data formats, enabling seamless implementation of our algorithm in commercial 2D-SIM data processing.

An inherent limitation of all SIM algorithms is the reduced reconstruction quality when processing raw data with degraded illumination pattern quality, due to the shared theoretical basis of structured illumination and demodulation. Such degradation can result, for example, from system imperfections, coverslip misalignment, low signal-to-noise ratio, or ultrafast motion blur. In the case of Lock-in-SIM, this issue may be further exacerbated due to its increased reliance on the accurate extraction and utilization of a clean grating illumination pattern. By precisely controlling data acquisition quality[25] and further integrating advancements such as adaptive optics[62,63], illumination field flatting[64], and rationalized pattern-aware deep learning denoising[27], the high-fidelity background suppression capability of Lock-in-SIM can be expected to be maintained even under highly challenging experimental conditions.

The compelling, high-quality imaging results of Lock-in-SIM are achieved by exploiting the modulation difference between the in-focused signal and background, and then utilizing lock-in demodulation to separate them. As a classic technique for extracting weak electrical signals from high background noise, lock-in detection has also demonstrated its capability of removing background in fluorescence image processing in recent years[48,49]. Compared to these methods, our lock-in demodulation strategy features prominent advantages. First, current optical lock-in detection methods require either specialized hardware[49] or blinking fluorescent dyes[48] to generate a periodic reference signal, while Lock-in-SIM is based on the intrinsic modulation characteristics of any existing SIM system and thus has no additional experimental requirements. Furthermore, Lock-in-SIM separates in-focused signal from the background through direct, fast, and parameter-free AC-DC demodulation, without sacrificing the nine-image speed benefit of SIM, while tens to hundreds of raw images are necessary for established correlation- or Fourier transform-based optical lock-in demodulation[48,49].

We demonstrate the methodological effectiveness and biological implications of our Lock-in-SIM method across a variety of challenging imaging scenarios, especially for densely packed intraorganellar ultrastructures whose sizes are near the resolution of SIM. Benefiting from the excellent background-removal ability of Lock-in-SIM, we were able to clearly capture the rapid morphological changes and interactions of the ER matrix, lysosomes, and mitochondrial inner membranes. We first found that lysosomes could control both the ER tubular remodeling and the reshaping of the ER matrix. Furthermore, we observed ER-mediated lysosome tubule fission and elucidated the influence of lysosome tubule length on the fission position. Finally, we discovered that mitochondria could regulate the fission of other mitochondria and revealed the specific molecular mechanisms involved. These investigations and findings substantially support and expand the existing dynamic and interaction models of organelles[30,43–50] and enhance our understanding of fundamental cellular processes and disease pathogenesis.

In conclusion, by improving imaging quality to approach the optimal SIM performance through Lock-in-SIM reconstruction, our research underscores the importance of background-free reconstruction of easily obscured ultrastructures, thereby inspiring novel biological discoveries.

## Methods

### Cell lines

U2OS cells (Human osteosarcoma cell line) were cultured in McCoy's 5A medium (Gibco) supplemented with 10% (v/v) fetal bovine serum (FBS; Gibco); COS-7 and mouse C127 epithelial cells were cultured in high-glucose Dulbecco's modified Eagle's medium (Gibco), supplemented with 10% FBS and 1% penicillin-streptomycin (Beyotime). Cell lines were purchased from Procell Life Science & Technology Co., Ltd. or the American Type Culture Collection. All cells were cultured at 37 °C and 5% $CO_2$.

### Sample preparation

For low-background bead sample preparation, 100 nm diameter yellow-green fluorescent microspheres (ThermoFisher) were diluted 1:2000 in phosphate-buffered saline (PBS) solution. For high-background bead sample preparation, the beads were diluted 1:2000 in PBS solution containing 2 μM Fluorescein 5-isothiocyanate (FITC, ThermoFisher). Bead solutions were then added to 35-mm glass-bottom dishes (Cellvis) for imaging.

For live-cell microtubule labeling in U2OS cells, we followed a well-established protocol[65]. Briefly, cells were co-incubated with mixLPV and 5 μM Tubulin-Atto 488 at 37 °C for 1 h. The cells were then washed three times with culture medium and incubated for another hour at 37 °C and 5% $CO_2$. Before imaging, the medium was replaced with phenol red-free McCoy's 5A medium.

For live-cell inner mitochondrial membrane labeling in U2OS cells, cells were seeded in a 35-mm glass-bottom dish at 37 °C with 5% $CO_2$ for 12–16 h. Then, the cells were co-incubated with 200 nM PKmito DEEP RED or 500 nM MitoTracker Green FM for approximately 30 min. The PKmito DEEP RED or MitoTracker Green FM dyes were then removed, and the cells were washed three times with PBS. Afterward, the cells were cultured for another 30 min at 37 °C and 5% $CO_2$. Before imaging, the medium was replaced with phenol red-free McCoy's 5A medium.

To label ER, mitochondria, lysosomes, and Drp1 for live-cell imaging, U2OS cells were first seeded in a 24-well plate. After 12 h, cells were transfected with 300 ng of the indicated plasmid (Sec61β-EGFP for ER, Tomm20-EGFP for mitochondria, Rab7-EGFP for lysosome, or mCherry-Drp1 for Drp1) using 1 μl of Lipofectamine 2000 in OPTI-MEM medium. Twenty-four hours post-transfection, the cells were detached with 0.25% trypsin-EDTA, re-seeded onto 35-mm glass-bottom dishes, and cultured for an additional 24 h at 37 °C and 5% $CO_2$ prior to imaging.

For live-cell (Sec61β-DY549) ER and (Rab7-AF488) lysosomes dual-color labeling, U2OS cells were seeded in a 24-well plate. After 12 h, the cells were transfected with OPTI-MEM medium (Invitrogen) containing 1 μl of Lipofectamine 2000 (Invitrogen) and 300 ng of the indicated plasmids (Sec61β-SNAP for ER and Rab7-Halo for lysosomes). Twenty-four hours post-transfection, the cells were detached with 0.25% trypsin-EDTA (Gibco), re-seeded onto 35-mm glass-bottom dishes (Cellvis), and cultured for an additional 24 h at 37 °C and 5% $CO_2$. For fluorescent labeling, the cells were then co-incubated with mixLPV along with 5 μM SNAP-DY549 and 5 μM Halo-AF488 for 1 h at 37 °C. Following incubation, the cells were washed three times with culture medium and then rested for another hour. Finally, prior to imaging, the medium was replaced with phenol red-free McCoy's 5A medium. All incubations were performed at 37 °C with 5% $CO_2$.

For mitochondria and mitochondrial-derived vesicles labeling, U2OS cells were seeded in 35-mm glass-bottom dishes (Cellvis). After 12 h, the cells were transfected with OPTI-MEM medium (Invitrogen) containing 1 μl of Lipofectamine 2000 (Invitrogen) and 300 ng Tomm20-Halo. Twenty-four hours post-transfection, the cells were then co-incubated with mixLPV along with 5 μM Halo-AF488 for 1 h at 37 °C. Following incubation, the cells were washed three times with culture medium and then rested for another hour. Finally, prior to imaging, cells were fixed with 2% glutaraldehyde.

For fixed-cell EdU replication labeling, mouse C127 epithelial cells were grown on #1.5H high precision coverslips and incubated for 15 min with 10 μM 5-ethynyl-2′-deoxyuridine before fixation with 3% formaldehyde/0.1% Triton X-100/PBS for 10 min. After washing cells

with PBS/0.05% Tween (PBST), cells were blocked with BlockAid (ThermoFisher) before click detection of incorporated EdU with AF488-azide using the Click-iT kit (ThermoFisher) and mounting in EverBrite medium (Biotium) following thorough washing with PBST.

For fixed-cell microtubule labeling, U2OS cells were seeded in a 35-mm glass-bottom dish at 37 °C with 5% CO$_2$ for 12–16 h. Before fixation, the cells were washed three times with PBS, followed by treatment with fixative buffer (3% paraformaldehyde, 0.1% glutaraldehyde, and 0.2% Triton X-100) for 15 min. The cells were then incubated with 0.2% Triton X-100 for another 15 min and blocked with blocking buffer (3% bovine serum albumin and 0.05% Triton X-100) for 20 min at room temperature. Following this, the cells were incubated overnight at 4 °C with anti-alpha-tubulin antibody (ab7291, Abcam, 1:500 dilution), after which the primary antibody was removed, and the cells were washed twice with PBS. The cells were then incubated with Alexa Fluor 488-conjugated secondary antibody (ab150113, Abcam, 1:600 dilution) for another 2 h at room temperature. Finally, the antibody was removed, and the cells were washed three times with PBS before imaging.

For fixed-cell mitochondrial or lysosomal labeling, the pre-sample preparation steps were the same as those described above for microtubule sample preparation. Specifically, the cells were incubated overnight at 4 °C with either an anti-Tomm20 antibody (ab232589, Abcam, 1:500 dilution) or an anti-Rab7 antibody (ab137029, Abcam, 1:500 dilution). The primary antibody was then carefully removed, and the cells were washed twice with PBS. Afterward, the cells were incubated with Alexa Fluor 488-conjugated secondary antibody (Abcam, 1:600 dilution) for 2 h at room temperature. Finally, the cells were washed three times with PBS.

To label the ER in fixed cells, U2OS cells were initially transfected with Sec61β-EGFP using Lipofectamine 2000 according to the standard protocol. Subsequently, the cells were cultured at 37 °C with 5% CO$_2$ for 24 h. Prior to imaging, the cells were fixed with 2% glutaraldehyde for 20 min.

For fixed-cell two-color labeling of microtubules and lysosomes, the cells were fixed with 2% glutaraldehyde for 20 min. The cells were then incubated with 0.2% Triton X-100 for another 15 min and blocked with blocking buffer (following the above microtubule protocol) for 20 min at room temperature. Following this, the cells were incubated with mouse anti-alpha-tubulin antibody (ab7291, Abcam, 1:500 dilution) and rabbit anti-Rab7 antibody (ab137029, Abcam, 1:500 dilution) overnight at 4 °C, after which the primary antibodies were removed, and the cells were washed twice with PBS. The cells were then incubated with goat anti-mouse Alexa Fluor 488 antibody and goat anti-rabbit Cy3 antibody (ab6939, Abcam, 1:800 dilution) for another 2 h at room temperature. Finally, the antibodies were removed, and the cells were washed three times with PBS before imaging.

**Experimental SIM data acquisition.** The SIM datasets were mainly acquired using a custom-built high-speed SIM system equipped with 2D-SIM, TIRF-SIM, and 3D-SIM imaging modalities[66,67], and commercial SIM systems (GE DeltaVision OMX SR; CSR Biotech, HIS-SIM; and NanoInsights-Tec, Multi-SIM). The main hardware components of the custom-built system included the following: multi-line lasers (488 nm, 561 nm, 594 nm, and 637 nm); scanning galvanometer sets (Cambridge Technology, CTI 8310k) for the incident angle and azimuth changes; electro-optic modulators (Thorlabs, EO-PM-NR-C4) for fast phase shifting; pizza-type half-wave plate (Union Optics, WPA2210-450-650) for polarization control to maintain high pattern contrast; objective lens (Nikon, 100×/1.49 oil) for laser incidence and fluorescence collection; and sCMOS camera (Hamamatsu, C13440-20CU). The pixel size of the raw image was 65 nm. The typical exposure times for the fixed-cell and live-cell experiments were 50 ms and 20–30 ms, respectively. Time intervals of 2–3 s could be set for longer-duration live-cell acquisition. More details, including the other hardware

components, software control, and resolution validation regarding our galvanometer-based SIM system, can be found in our published works[66,67]. In addition, the open-source datasets (Supplementary Figs. 7–13) were also used for comparison and validation, demonstrating the universality and advances of our Lock-in-SIM approach.

For most experiments, single slices were recorded in 2D-SIM mode, in which the data were acquired by manually focusing on the plane with the highest image contrast, i.e., the plane with the most structure and least background. For the optical sectioning 2D-SIM experiments (Fig. 3, Supplementary Figs. 20–22), the sample thickness and focus position are labeled in the corresponding figures. For comparison, the optical sectioning 2D-SIM data were typically acquired with the same axial step size as the 3D-SIM data (125 nm). The number of steps depended on the imaging depth. Note that the axial step size can also be set to ~250 nm to improve the acquisition speed of the whole stack, because although our Lock-in-SIM method minimizes the background and improves the optical sectioning ability of standard 2D-SIM, its axial resolution is still limited to the 2D-SIM level (Supplementary Fig. 21).

**Simulated and synthetic SIM data generation.** The simulated data in Fig. 1 and Supplementary Fig. 3 were generated in MATLAB. The ground-truth structures of the crossed filaments and radiative lines were multiplied by the sinusoidal SIM pattern and convoluted with the PSF. The simulation parameters were set to match the experimental conditions: $\lambda = 488$ nm, numerical aperture = 1.49, and pixel size = 65 nm. Gaussian noise with different standard deviations and contrast with different levels could be added for robustness tests. The synthetic data in Supplementary Fig. 5 were generated by superimposing the widefield images on the raw SIM images at different levels.

**SBR calculation.** To evaluate the background-removing capability of each algorithm quantitatively, we calculated SBR values with the following definition[68]:

$$SBR = 10 \log_{10} \frac{\max(\text{signal}) - \text{mean}(\text{background})}{\text{s.d.}(\text{background})} \quad (4)$$

where max(signal), mean(background), and s.d.(background) represent the maximum intensity of the in-focused signal, the average intensity of the background, and the standard deviation of the background, respectively. Note that the background value used in the SBR calculation was determined from all pixels without a structural signal. Therefore, the SBR includes the contribution of noise as well.

**Orientation analysis.** The quantitative, pixel-wise orientation values ($\theta$) of fibrous structures, such as microtubules and mitochondria, were calculated with a weighted vector summation algorithm[51,52] consisting of five steps. First, a typical window size of $7 \times 7$ was applied to the image. Forty-eight vectors were defined from the central pixel to the surrounding pixels. Second, the first weight factor $F_1$ for each vector was calculated as the inverse of its length. Third, the second weight factor $F_2$ for each vector was calculated as the intensity difference among the three pixels along the vector:

$$F_2 = \sqrt{\frac{1}{3} - \sqrt{\frac{1}{2}\sum_{i=1}^{3}(\text{Intensity}_i - \overline{\text{Intensity}})^2}} \quad (5)$$

Fourth, the two weight factors were summed to obtain the average direction of the central pixel. Finally, the summed direction was corrected with prior before converting the orientation angle to be between 0° and 180°.

**Image processing and visualization.** Wiener-SIM reconstructions were performed using a custom software based on standard Wiener-SIM developed by Gustafsson[6]. Fair-SIM reconstructions were performed using the open-access Fair-SIM Fiji/ImageJ plugin. WLR-SIM reconstructions were performed by the original authors. Open-SIM, Hessian-SIM, IM-SIM, SP-SIM, HiFi-SIM, JSFR-SIM, PCA-SIM, BF-SIM, Direct-SIM, and Flex-SIM reconstructions were performed using corresponding open-access MATLAB codes with their default or recommended parameters (Supplementary Table 2). Some of the reconstructed results from the existing methods with their own datasets were, where available, either directly picked from their open-access files or reproduced with the provided parameter-optimized codes, such as BF-SIM, Direct-SIM, and Flex-SIM. The processing speed of Lock-in-SIM (~8.68 s for a single-frame 512 × 512 2D-SIM dataset on an Intel i5-13500 2.50 GHz CPU with 64 GB RAM) is at a moderate level among the tested algorithms (Supplementary Table 2), which can be further improved through hardware optimization, software integration, and GPU acceleration to facilitate the real-time, high-SBR live-cell imaging[69]. Because of the possible intensity variations among different frames, either optical-sectioning z stack or time-lapse t stack, we may correct the intensity variation after reconstruction with the ImageJ plugin (https://imagej.net/plugins/bleach-correction). SIM images might also be adjusted and smoothed with a 2D Gaussian filter of 0.5–0.6 pixels for visualization purposes. 3D volumetric data were mainly rendered and visualized with the ImageJ Volume Viewer plugin and Imaris software.

**Statistical analysis.** The Fourier domain evaluation in Supplementary Fig. 14 was performed using the SIMcheck Fiji/ImageJ plugin[70]. Decorrelation analysis[71] and full-width at half maximum measurement were used to assess spatial resolution. The statistical data in Fig. 4l and Supplementary Figs. 6 and 25 were plotted in the box-and-whisker format. The middle line in the box corresponds to the mean value. The upper and lower boundary lines correspond to the 15th and 75th percentiles, respectively. The whiskers represent the min and max values. Two-tailed $t$-tests were applied to compare the groups. $P$ values ≥ 0.05 were considered not different and labeled ns; $P$ values < 0.05 were considered significantly different and labeled *; $P$ values < 0.0001 were considered extremely significantly different and labeled ****.

### Reporting summary
Further information on research design is available in the Nature Portfolio Reporting Summary linked to this article.

## Data availability
The source data that support the findings of this study are publicly available at https://figshare.com/articles/figure/Raw_datasets/26130994. Source data are provided with this paper.

## Code availability
The custom MATLAB codes, GUI, Fiji/ImageJ plugin, and executable software of Lock-in-SIM are available at https://github.com/WenjieLab/Lock-in-SIM[72].

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

## Acknowledgements

The authors thank M. Müller (Bielefeld University) for help with Fiji/ImageJ plugin development. The authors thank M. Shaw (National Physical Laboratory) and Y. Chen (Zhejiang University) for help with algorithm testing. The authors thank J. C. Thiele (University of Oxford) and J. Wang (University of Oxford) for fruitful technical discussions. The authors thank X. Li (Shanghai Jiaotong University) for sample preparation

assistance. The authors apologize for any missing algorithm in their method comparison. The authors acknowledge and thank all SIM developers and users for their significant contributions to the flourishing SIM community. The authors thank the Micron Bioimaging Facility (University of Oxford), Optical Bioimaging Core Facility (Wuhan National Laboratory for Optoelectronics, Huazhong University of Science and Technology), and Core Facilities (Zhejiang University School of Medicine) for experimental assistance. W.L. acknowledges support by the European Commission under the Marie Skłodowska-Curie Programme No. 101202243. L.S. acknowledges support by the Wellcome Trust Strategic Award (107457/Z/15/Z) to fund the Micron Oxford Advanced Bioimaging Unit, and the European Union's Horizon 2020 research and innovation program under the Marie Skłodowska-Curie Grant Agreement No. 766181. Y.-H.Z. acknowledges support by the National Key Research and Development Program of China (2022YFC3401100) and the National Natural Science Foundation of China (92354305). Y.C. acknowledges support by the Natural Science Foundation of Zhejiang Province (LY23F050010) and the National Key Research and Development Program of China (2021YFF0700302). M.Z. acknowledges support by the National Natural Science Foundation of China (32201132) and China Postdoctoral Science Foundation-funded project (BX20220125 and 2022M711257). C.K. and Z.L. acknowledge support by the National Natural Science Foundation of China (62125504 and 62275232).

## Author contributions

W.L., L.S., Y.-H.Z. and Y.C. supervised the project. W.L., W.Z., and X.X. developed the algorithm and wrote the custom code. L.S., M.Z., Y.C., Y.-H.Z., J.Z., S.Q., Z.L., X.Z., D.Z., J.D., X.L. and C.K. helped improve and test the algorithm. W.L., M.Z., L.S., Y.-H.Z., Y.C. and J.Z. designed and performed fixed-cell experiments. W.L., M.Z., L.S., Y.-H.Z., Y.C., Q.F., W.Y. and Y.W. designed and performed live-cell experiments. W.L., M.Z., L.S., Y.-H.Z. and Y.C. analyzed data. W.L. and M.Z. composed the figures and movies with input from L.S., Y.-H.Z., Y.C., W.Z. and J.Z., and all the other authors. W.L. and M.Z. wrote the manuscript with input from L.S., Y.-H.Z., Y.C., W.Z., J.Z. and all the other authors. All authors discussed the results and commented on the manuscript.

## Competing interests

The authors declare no competing interests.
