## [Transparent Peer Review file · Nature Communications]

Visualizing intraorganellar ultrastructures, dynamics, and interactions with open-access background-free Lock-in-SIM

Corresponding Author: Dr Wenjie Liu

Version 0:

Reviewer comments:

Reviewer #1

(Remarks to the Author)

This study presents Lock-in-SIM, a novel background suppression method based on a spatial domain model to address out-of-focus signal artifacts in 2D Super-resolution Structured Illumination Microscopy (2D-SR-SIM). By leveraging the modulation properties of SIM, the authors propose an effective signal extraction framework that eliminates background without requiring additional hardware modifications. The method significantly enhances the signal-to-background ratio (SBR) and resolution while preserving the integrity of super-resolution structures. This approach is innovative and practical, and experimental results comprehensively validate Lock-in-SIM across various biological samples, imaging conditions, and datasets. Notably, the method demonstrates robustness in high-background environments while maintaining structural accuracy, making it particularly useful for visualizing and quantitatively analyzing intraorganellar ultrastructures.

The authors provide a well-documented algorithm implementation, along with functional and accessible software tools, including a MATLAB GUI, a Fiji/ImageJ plugin, and an executable program, making it easy to reproduce and apply.

Overall, the study is well-motivated, technically sound, and experimentally thorough, with significant potential for broad application in super-resolution imaging. However, there are a few aspects that require further clarification to enhance the completeness and impact of the manuscript.

Approximation in Lock-in-SIM

[SI Line 96-97] In Supplementary Note 1 the final step of Lock-in demodulation does not explicitly include the modulation depth m . Clarification on the typical range of m in practical applications and why it can be omitted in this step would be solid.

Comparison with Other Methods

While the paper quantitatively compares Lock-in-SIM with 13 existing methods, additional details on parameter choices and computational efficiency would help further clarify its advantages and potential limitations. Specifically, it would be useful to explain how parameters were selected for each method to ensure a fair comparison, as different settings can significantly impact reconstruction quality. Additionally, a discussion of computational efficiency—such as processing time and resource requirements—would provide valuable insight into the practical applicability of Lock-in-SIM, particularly for live-cell imaging where speed and real-time processing are important considerations.

Clarification on PSF Modeling in BF-SIM

[SI Line 173-179] In Supplementary Note 3, the discussion of BF-SIM states that it models the PSF as a 3D distribution, but the description of the challenges involved could be clarified. BF-SIM typically generates its PSF based on system parameters such as the objective lens properties, rather than requiring experimental measurement for each specific condition. While mismatches between the modeled and actual PSF may still affect reconstruction quality, the current wording could be interpreted as suggesting that BF-SIM requires direct experimental PSF measurement or extensive per-condition adjustments. The authors may consider refining this discussion to more precisely describe the PSF modeling process in BF-SIM and the extent to which it depends on experimental conditions.

Minor Formatting and Writing Issues

Below are a few minor issues related to formatting, writing, and figure legends that could be addressed for clarity and consistency:

1. [Line 185-187] The expression and crossref seems a bit obscure. In supplementary Fig. 2, only simulated filaments are presented.
2. [Line 265-266] Should the corresponding figure be Extend Data Fig. 4?
3. [Fig 3 caption] In Line 902, “b(orange histogram)” should be corrected to “c(orange histogram)”.
4. [Fig 4b, h, Supplementary Fig. 16b] The color of the legend indicating orientation should have an angular gradient, not a linear gradient.
5. [Supplementary Fig.4 caption] In Line 243, “without (upper left) and with (bottom right) Lock-in filtering” should be corrected to “without (bottom right) and with (upper left) Lock-in filtering”.

(Remarks on code availability)

Reviewer #2

(Remarks to the Author)

Liu et al. present a new computational approach for structured illumination microscopy image reconstruction that takes advantage of several tricks from conventional signal processing. They show that by using the input grating patterns as a “lock-in” signal, they can correct the background appropriately and generate some really remarkably high-quality SIM reconstructions. They go on to demonstrate using a battery of conventional targets for SIM that their system provides an equal-or-better reconstruction than nearly any other SIM reconstruction approach currently available, and even create a few nice packaged versions of the software to allow the free distribution of the technique.

In general, I am extremely enthusiastic about this paper. It marks the culmination of many years of work by many groups to generate a generalizable, well-rounded reconstruction approach that can be broadly distributed. They do sufficient work to demonstrate its utility across various samples, and I am very pleased to see the Fiji plugin and MATLAB GUI designed to bring it to the average user. I am particularly impressed on how well they have represented all of the competing techniques throughout the paper, allowing a new user to really see for themselves where and how this technique will help them. I have only one major concern and a few minor points before strongly recommending this for publication, I have detailed them below:

Major Point:

One presumable weakness of this approach is that it requires the ability to “lock-in” on the modulated (excitation) signal that comes from the patterned illumination. The authors show lots of examples of structures with one- or two-dimensional continuity, which makes this clear. I could not find any examples where the authors test this on small, sub-diffraction limited puncta, especially those moving very quickly (fast enough to blur during 2D image acquisition). Would this be predicted to be problematic? I think it would be worth demonstrating whether this was true or not—either as a caution or as an explicit endorsement of using this technique for the numerous small puncta experiments common in biology (e.g., mitochondrial derived vesicles, plasma membrane microdomains, etc.).

The second implication of this is that the fidelity of Lock-In SIM reconstruction is going to be dependent on the ability of the scope to produce a clean grating illumination pattern—incomplete or uneven interference patterns (as can happen when coverslips are not flat on the objective, etc.) will still cause the reconstruction to amplify artifacts (as do most of the other reconstruction algorithms, of course). Thus, I think the authors should be clear that Lock-In SIM is not an “artifact-free” approach, per se, but rather one that reduces background and improves about the reconstruction if the data is sufficiently high quality. This will be especially important for users that generate the SIM pattern by direct interferometry such as light sheet users, since slight misalignments in these systems can render the grating pattern difficult to fit. I have tried the Lock-In SIM on some data from my own group, and I notice that when the input data is clean, it works amazingly, but when the data has issues (poor interference fringes as a result of misaligned coverslip, in my case), and I do note erroneous breaking of continuous structures and amplification of noise in dark regions to generate artifactual puncta. To be clear, this is not a flaw with the technique, but I think the authors could explicitly state these limitations, at least in the supplement for experts running core facilities to watch for in their users’ data.

Minor Points:

1. In Extended Data Fig. 1, the authors label the bottom of the images “Nucleus.” Are the authors actually sure this is where the nucleus is (i.e. – they have some kind of DNA signal in another channel?). I ask because this is a very unusual ER structure to have under the nucleus, and looks much more like the dense “perinuclear” ER structures that happen around its edges. If they aren’t sure, I would label it more generically as “perinuclear” to avoid an allergic response from someone who studies the nuclear envelope (which does not seem to be visible here).
2. Can the authors make some quantitative estimates from their data about the z-resolution (or effective focal depth?) in their optical sectioning experiments? They demonstrate it is improved compared to 2D SIM, but it would be nice to have a sense of to what degree or at least order of magnitude. The results on the ER in Extended Data Fig. 1 suggest a dramatic improvement in reducing the parts of the organelle sectioned, it would be nice to know how much z-depth that corresponds to.

(Remarks on code availability)

I have tested both the Fiji plugin and the MATLAB code on approximately 50 images collected by my own group from different SIM implementations. I'd say on 90+% of them, it worked very well. I note it did introduce some artifacts and issues if the input data quality was poor (especially when misalignment of the system caused the underlying SIM illumination pattern to be difficult to make out). All SIM reconstruction patterns have issues with this kind of data, however, so I don't see this as a substantial flaw—but I note it existed and in some cases was exacerbated in the Lock-In SIM compared to simpler Weiner

filtered SIM reconstructions.

Version 1:

Reviewer comments:

Reviewer #1

(Remarks to the Author)

In this revised version of the manuscript entitled "Visualizing intraorganellar ultrastructures, dynamics, and interactions with open-access background-free Lock-in-SIM", the authors have provided thoughtful and effective responses to the comments raised in the first round of review. The revisions have improved both the technical clarity and the overall presentation of the work.

The clarification regarding the modulation depth m has effectively removed unnecessary ambiguity in the description of the Lock-in demodulation process. Similarly, the revised explanation of PSF modeling in BF-SIM has clarified that it relies on system parameters rather than direct experimental measurements, while still noting the potential impact of PSF mismatch. The additional details and discussions added to the comparison with other methods, particularly concerning parameter selection and computational efficiency, also contribute to a more balanced and informative evaluation of the proposed technique.

Minor formatting and figure-related issues mentioned previously have largely been resolved. The authors have also amended other minor errors missed in the initial review. The overall presentation of the manuscript is now clearer and more consistent.

In conclusion, the authors have sufficiently addressed the points raised during the first review round, and I have no further concerns. I think the manuscript is now in good shape for publication, pending any final editorial checks.

(Remarks on code availability)

The code is complete, clear, and well-documented. Using the data provided by the authors, I successfully reproduced all results which match those reported in the text. This is further facilitated by their accessible software tools, including the MATLAB GUI, Fiji/ImageJ plugin, and standalone executable, making both validation and application straightforward.

Reviewer #2

(Remarks to the Author)

The authors have exhaustively addressed all of my concerns, I particularly appreciate the attention to detail with the extra supplemental figures--very thoroughly done and well explained. I do not have any remaining concerns, I hope to see the paper in press soon.

(Remarks on code availability)

This was all evaluated and found to be outstanding in the previous round of reviews.

Reviewer #1:

Comment: This study presents Lock-in-SIM, a novel background suppression method based on a spatial domain model to address out-of-focus signal artifacts in 2D Super-resolution Structured Illumination Microscopy (2D-SR-SIM). By leveraging the modulation properties of SIM, the authors propose an effective signal extraction framework that eliminates background without requiring additional hardware modifications. The method significantly enhances the signal-to-background ratio (SBR) and resolution while preserving the integrity of super-resolution structures. This approach is innovative and practical, and experimental results comprehensively validate Lock-in-SIM across various biological samples, imaging conditions, and datasets. Notably, the method demonstrates robustness in high-background environments while maintaining structural accuracy, making it particularly useful for visualizing and quantitatively analyzing intraorganellar ultrastructures.

The authors provide a well-documented algorithm implementation, along with functional and accessible software tools, including a MATLAB GUI, a Fiji/ImageJ plugin, and an executable program, making it easy to reproduce and apply.

Overall, the study is well-motivated, technically sound, and experimentally thorough, with significant potential for broad application in super-resolution imaging. However, there are a few aspects that require further clarification to enhance the completeness and impact of the manuscript.

Response: We greatly appreciate the reviewer for the high recognition of our work, as well as for the careful reading and positive, constructive feedback to improve the manuscript. In the following, we give our responses point by point to the reviewer's comments. The revised portions are marked in blue in the revised manuscript.

Comment: Approximation in Lock-in-SIM

[SI Line 96-97] In Supplementary Note 1 the final step of Lock-in demodulation does not explicitly include the modulation depth m . Clarification on the typical range of m in practical applications and why it can be omitted in this step would be solid.

Response: We thank the reviewer for the insightful comment. In the algorithm design, we tested our algorithm both with and without including modulation depth during the *Step 2 Lock-in demodulation*, and found no obvious difference in the results. Therefore, we chose to omit modulation depth in this step to simplify the processing. Note that the modulation depth is still included and utilized in the subsequent *Step 3 Super-resolution reconstruction*. The comprehensive comparison presented in the manuscript

further validates the effectiveness of this processing strategy. Lock-in-SIM consistently achieves high reconstruction quality across a wide range of experimental conditions.

To clarify this, we have added the explanation in the revised supplementary material:

Revision:

(Supplementary Note 1, Page 4, Para. 2)

“The modulation depth m is omitted at this step to simplify the processing, without compromising the reconstruction quality.”

Comment: Comparison with Other Methods

While the paper quantitatively compares Lock-in-SIM with 13 existing methods, additional details on parameter choices and computational efficiency would help further clarify its advantages and potential limitations. Specifically, it would be useful to explain how parameters were selected for each method to ensure a fair comparison, as different settings can significantly impact reconstruction quality. Additionally, a discussion of computational efficiency—such as processing time and resource requirements—would provide valuable insight into the practical applicability of Lock-in-SIM, particularly for live-cell imaging where speed and real-time processing are important considerations.

Response: We thank the reviewer for this valuable suggestion and fully acknowledge the importance of providing these details. We have now added explanations of the parameter selection and have tested and discussed the computational speed. Please refer to the revised content below:

Revision:

(Methods, Image processing and visualization, Page 27, Line 747-759)

“**Image processing and visualization.** Wiener-SIM reconstructions were performed using a custom software based on standard Wiener-SIM developed by Gustafsson⁶. Fair-SIM reconstructions were performed using open-access Fair-SIM Fiji/ImageJ plugin. WLR-SIM reconstructions were performed by the original authors. Open-SIM, Hessian-SIM, IM-SIM, SP-SIM, HiFi-SIM, JSFR-SIM, PCA-SIM, BF-SIM, Direct-SIM, and Flex-SIM reconstructions were performed using corresponding open-access MATLAB codes with their default or recommended parameters (Supplementary Table 2). Some of the reconstructed results from the existing methods with their own datasets were, where available, either directly picked from their open-access files or reproduced with the provided parameter-optimized codes, such as BF-SIM, Direct-SIM, and Flex-SIM. The processing speed of Lock-in-SIM (~8.68 s for a single-frame 512×512 2D-SIM dataset on an Intel i5-13500 2.50 GHz CPU with 64 GB RAM) is at

a moderate level among the tested algorithms (Supplementary Table 2), which can be further improved through hardware optimization, software integration, and GPU acceleration to facilitate the real-time, high-SBR live-cell imaging⁷⁴.”

74. Xu, F. et al. Real-time reconstruction using electro-optics modulator-based structured illumination microscopy. *Opt. Express* **30**, 13238-13251 (2022).

Supplementary Table 2. Reconstruction parameters and speed of the algorithms.

	Parameters setting	Reconstruction speed (s)^a
WLR-SIM	β^b	
Open-SIM	No parameter setting	44.63
Fair-SIM	Wiener parameter=0.05, APO cutoff=2, APO bend=0.9	γ^c
Hessian-SIM	Wiener parameter=2, Theta Ratio=1:1:1, Mu=150, Sigma=1	γ^d
IM-SIM	wiener_factor=0.05	4.02
SP-SIM	wienerFactor=0.05	1.60
HiFi-SIM	attStrength=0.6-0.9, ApoFWHM=0.5, $\beta=1$, w1=1.2	9.23
JSFR-SIM	WIENER PARAMETER=0.5, AMP=1, SIGMA=1.5	γ^c
PCA-SIM	sub_optimization=1, Filter_size=11, Mask_size=3	3.79
BF-SIM	Camera Offset=100, Wiener para=1.5	44.88
Direct-SIM	$\alpha=0.8$, $\beta=0.5$, AttStr.=0.3, FilterFWHM=0.5	5.23
Flex-SIM	params.OTFAttStr=0.999, params.OTFAttwidth=0.3, params.apodize = 0, params.sepOrr = 0, params.padSz=10, params.mu = 7e-6, params.regType=1, params.maxIt = 100, params.stepTol = 1e-3	43.32 ^e
Lock-in-SIM	Lock-in parameter=0.5-0.8	8.68

^a The speed was tested on a Windows 10 desktop equipped with a 13th Gen Intel(R) Core(TM) i5-13500 2.50 GHz CPU and 64 GB RAM, using a single-frame 512*512 2D-SIM raw dataset.

^b The WLR-SIM reconstruction was performed and shared by the original authors.

^c Reconstructed speeds for Fair-SIM and JSFR-SIM were not reported here due to their separated multi-step processing. Readers can refer to the original publications^{11,17} for more details. Notably, JSFR-SIM is expected to offer highly increased speed due to the accelerated spatial-domain reconstruction.

^d The reconstructed speed of Hessian-SIM was not reported here due to its multi-frame input requirement.

^e The Flex-SIM speed was tested on a MacBook Pro laptop equipped with Apple M1 Pro and 16 GB RAM, due to the incomplete running on the Windows computer.

Comment: Clarification on PSF Modeling in BF-SIM

[SI Line 173-179] In Supplementary Note 3, the discussion of BF-SIM states that it models the PSF as a 3D distribution, but the description of the challenges involved could be clarified. BF-SIM typically generates its PSF based on system parameters such as the objective lens properties, rather than requiring experimental measurement for each specific condition. While mismatches between the modeled and actual PSF may still affect reconstruction quality, the current wording could be interpreted as suggesting that BF-SIM requires direct experimental PSF measurement or extensive per-condition adjustments. The authors may consider refining this discussion to more precisely describe the PSF modeling process in BF-SIM and the extent to which it depends on experimental conditions.

Response: We appreciate the reviewer for highlighting the misunderstanding caused by our inappropriate wording. BF-SIM indeed used simulated PSF for reconstruction. We have refined the corresponding discussion to improve clarity:

Revision:

(Supplementary Note 3, Page 7, Para. 3)

“By generating a simulated 3D PSF based on system parameters including the immersion refractive index, wavelength, numerical aperture, and pixel size, and subsequently subtracting the out-of-focus PSF contribution, BF-SIM achieves effective background suppression. However, accurately modeling an actual 3D PSF in simulation to align with complicated experimental conditions is challenging. Any mismatch, for example due to optical aberrations, may compromise the reconstruction quality.”

Comment: Minor Formatting and Writing Issues

Below are a few minor issues related to formatting, writing, and figure legends that could be addressed for clarity and consistency:

1. [Line 185-187] The expression and crossref seems a bit obscure. In supplementary Fig. 2, only simulated filaments are presented.
2. [Line 265-266] Should the corresponding figure be Extend Data Fig. 4?
3. [Fig 3 caption] In Line 902, “b(orange histogram)” should be corrected to “c(orange histogram)”.
4. [Fig 4b, h, Supplementary Fig. 16b] The color of the legend indicating orientation should have an angular gradient, not a linear gradient.
5. [Supplementary Fig.4 caption] In Line 243, “without (upper left) and with (bottom right) Lock-in filtering” should be corrected to “without (bottom right) and with (upper

left) Lock-in filtering”.

Response: We thank the reviewer again for the meticulous reading and apologize for these confusions and mistakes. All the noted issues have been corrected, and we have also conducted multiple careful reviews to identify and correct other potential writing mistakes, as well as refine the overall clarity and expression.

For the reviewer’s convenience, these revisions have been appended below:

Revision:

1. (*Results, Lock-in-SIM provides optimal background suppression, image resolution, and fidelity, Page 7, Line 185-187*)

Original: “Applied to simulated, synthetic, **and** experimental bead datasets, Lock-in-SIM reconstructions proved highly robust against variations in noise (Supplementary Fig. 2) ...”

Revised: “Applied to simulated, synthetic, **or** experimental bead datasets, Lock-in-SIM reconstructions proved highly robust against variations in noise (Supplementary Fig. 2) ...”

2. (*Results, High-fidelity whole-cell optical sectioning reconstruction using Lock-in-SIM with standard 2D-SIM acquisition, Page 10, Line 267-268*)

Original: “Moreover, we benchmarked Lock-in-SIM with standard 3D-SIM on the same data (Extended Data Fig. **3** and Supplementary Video 8).”

Revised: “**Benchmarking Lock-in-SIM against** standard 3D-SIM **of the same cell** (Extended Data Fig. **4**, **Supplementary Fig. 18**, and Supplementary Video 8), ...”

3. (*Fig. 3 caption, Page 35, Line 908*)

Original: “Orientation distribution of the cell shown in **a** (green histogram) and in **b** (orange histogram).”

Revised: “Orientation distribution of the cell shown in **a** (green histogram) and in **c** (orange histogram).”

4. (*Fig. 4b, h, Supplementary Fig. 19b, Color bar of θ*)

5. (*Supplementary Fig. 4 caption, Page 11*)

Original: “Widefield image and super-resolution images reconstructed without (**upper left**) and with (**bottom right**) Lock-in filtering.”

Revised: “Widefield image and super-resolution images reconstructed without (bottom right) and with (upper left) Lock-in filtering.”

Other revision examples:

6. (*Introduction, Page 4, Line 105-107*)

Original: “This process is inherent in SIM imaging and **thus can** be applied to data acquired from existing SIM systems **and be integrated** into established postprocessing workflows, without additional hardware modifications and troublesome parameter finetuning.”

Revised: “This process is inherent in SIM imaging and **can therefore** be applied to data acquired from existing SIM systems, **allowing for seamless integration** into established post-processing workflows without additional hardware modifications and troublesome parameter fine-tuning.”

7. (*Fig. 3 caption, Page 35*)

Original: “Depth-coded maximum intensity projection of a 4.625- μm thick image stack of anti-Tomm20-labeled **mitochondrial** reconstructed using Wiener-SIM (left) and Lock-in-SIM (middle).”

Revised: “Depth-coded maximum intensity projection of a 4.625- μm thick image stack of anti-Tomm20-labeled **mitochondria** reconstructed using Wiener-SIM (left) and Lock-in-SIM (middle).”

8. (*Extended Data Fig. 10 caption, Page 51*)

Original: “Two-color time-lapse images of Tomm20-**EFGP**-labeled mitochondrial outer membrane (cyan) and mCherry-labeled Drp1 oligomers (red).”

Revised: “Two-color time-lapse images of Tomm20-**EGFP**-labeled mitochondrial outer membrane (cyan) and mCherry-labeled Drp1 oligomers (red).”

9. (*Supplementary Fig. 9 caption, Page 16*)

Original: “Reconstruction comparison of open-access **simulaed** and experimental data¹⁹ using different 2D-SIM algorithms.”

Revised: “Reconstruction comparison of open-access simulated and experimental data¹⁹ using different 2D-SIM algorithms.”

Reviewer #2:

Comment: Liu et al. present a new computational approach for structured illumination microscopy image reconstruction that takes advantage of several tricks from conventional signal processing. They show that by using the input grating patterns as a “lock-in” signal, they can correct the background appropriately and generate some really remarkably high-quality SIM reconstructions. They go on to demonstrate using a battery of conventional targets for SIM that their system provides a equal-or-better reconstruction than nearly any other SIM reconstruction approach currently available, and even create a few nice packaged versions of the software to allow the free distribution of the technique.

In general, I am extremely enthusiastic about this paper. It marks the culmination of many years of work by many groups to generate a generalizable, well-rounded reconstruction approach that can be broadly distributed. They do sufficient work to demonstrate its utility across various samples, and I am very pleased to see the Fiji plugin and MATLAB GUI designed to bring it to the average user. I am particularly impressed on how well they have represented all of the competing techniques throughout the paper, allowing a new user to really see for themselves where and how this technique will help them.

I have only one major concern and a few minor points before strongly recommending this for publication, I have detailed them below:

Response: We appreciate the reviewer for thorough reading, careful testing, and valuable feedback on our manuscript. We also greatly appreciate the reviewer’s high recognition of our work. We sincerely hope to make the meaningful contribution to the bioimaging community as highlighted by the reviewer. In the following, we give our responses point by point to the reviewer’s comments. The revised portions are marked in blue in the revised manuscript.

Comment: Major Point:

One presumable weakness of this approach is that it requires the ability to “lock-in” on the modulated (excitation) signal that comes from the patterned illumination. The authors show lots of examples of structures with one- or two-dimensional continuity, which makes this clear. I could not find any examples where the authors test this on small, sub-diffraction limited puncta, especially those moving very quickly (fast enough to blur during 2D image acquisition). Would this be predicted to be problematic? I think it would be worth demonstrating whether this was true or not—either as a caution or

as an explicit endorsement of using this technique for the numerous small puncta experiments common in biology (e.g., mitochondrial derived vesicles, plasma membrane microdomains, etc.).

Response: We appreciate the reviewer's thoughtful suggestion to strengthen the demonstration of our method's applicability and apologize for the lack of clarity in the previous presentation. Lock-in-SIM is indeed also suited for small puncta applications. In the original manuscript, we have shown the effective Lock-in-SIM reconstruction on biological clathrin-coated pit puncta (Supplementary Fig. 15), in addition to the common fluorescent bead experiments (Supplementary Figs. 3,5,8,9). To further enhance the puncta demonstration and provide an explicit endorsement, we conducted additional small puncta experiments, including mitochondrial-derived vesicles (Supplementary Fig. 16) and 5-ethynyl-2'-deoxyuridine (EdU) replication labeled nuclei (Supplementary Fig. 17). All together, these results highlight the reconstruction enhancement robustness of Lock-in-SIM across a variety of biologically structures.

We have incorporated these new datasets and added corresponding discussions into the revised manuscript. These revisions are also provided below for the reviewer's convenience:

Revision:

Supplementary Fig. 16. Reconstruction comparison of Tomm20-AF488-labeled mitochondria and mitochondrial-derived vesicles. a, Widefield image and corresponding super-resolution images reconstructed using Wiener-SIM and Lock-in-SIM. b,c Magnified images of the white boxed mitochondrial-derived vesicles in a. From top to bottom, Widefield, Wiener-SIM, and Lock-in-SIM. Scale bars, 3 μm (a) and 0.1 μm (b,c).

Supplementary Fig. 17. Optical-sectioning reconstruction comparison of EdU replication-labelled nuclei. **a-c**, Widefield images and corresponding super-resolution images reconstructed using Wiener-SIM and Lock-in-SIM of the sample at different sample depths. **d-f**, Magnified images of the white boxed regions in **a-c**. From top to bottom, Widefield, Wiener-SIM, and Lock-in-SIM. Scale bars, 3 μm (**a-c**) and 0.5 μm (**d-f**).

(Results, Lock-in-SIM provides optimal background suppression, image resolution, and fidelity, Page 7, Line 196-201)

“We next validated the superiority of Lock-in-SIM for different sample types with zero-, one-, two-, or three-dimensional continuity and varying imaging conditions using both open-access (Supplementary Figs. 6-12) and own experimental datasets (Fig.2, Extended data Figs. 2,3, Supplementary Figs. 13-18, and Supplementary Video 2), including the mitochondrial inner and outer membranes, microtubules, actin, lysosomes, clathrin-coated pits, mitochondrial-derived vesicles, and 5-ethynyl-2'-deoxyuridine (EdU) replication labeled nuclei.”

Comment: The second implication of this is that the fidelity of Lock-In SIM reconstruction is going to be dependent on the ability of the scope to produce a clean grating illumination pattern—incomplete or uneven interference patterns (as can happen when coverslips are not flat on the objective, etc.) will still cause the

reconstruction to amplify artifacts (as do most of the other reconstruction algorithms, of course). Thus, I think the authors should be clear that Lock-In SIM is not an “artifact-free” approach, per se, but rather one that reduces background and improves about the reconstruction if the data is sufficiently high quality. This will be especially important for users that generate the SIM pattern by direct interferometry such as light sheet users, since slight misalignments in these systems can render the grating pattern difficult to fit. I have tried the Lock-In SIM on some data from my own group, and I notice that when the input data is clean, it works amazingly, but when the data has issues (poor interference fringes as a result of misaligned coverslip, in my case), and I do note erroneous breaking of continuous structures and amplification of noise in dark regions to generate artifactual puncta. To be clear, this is not a flaw with the technique, but I think the authors could explicitly state these limitations, at least in the supplement for experts running core facilities to watch for in their users’ data.

Response: We are grateful for the reviewer’s insightful comments and fully acknowledge the reviewer’s precise understanding of the reduced reconstruction quality of the SIM technique in handling raw data with low illumination pattern quality. The issue might be further amplified in Lock-in-SIM due to its stronger dependence on accurately extracting and utilizing a clean grating illumination pattern. In response, we have expanded the Discussion section to explicitly remind this and to also discuss potential mitigation strategies. Please see below the revision:

Revision:

(Discussion, Page 14, Line 408-418)

“An inherent limitation of all SIM algorithms is the reduced reconstruction quality when processing raw data with degraded illumination pattern quality, due to the shared theoretical basis of structured illumination and demodulation. Such degradation can result from, for example, system imperfections, coverslip misalignment, low signal-to-noise ratio, or ultrafast motion blur. In the case of Lock-in-SIM, this issue may be further exacerbated due to its increased reliance on the accurate extraction and utilization of a clean grating illumination pattern. By precisely controlling data acquisition quality²⁵ and further integrating advancements such as adaptive optics^{62,63}, illumination field flattening⁶⁴, and rationalized pattern-aware deep learning denoising²⁷, the high-fidelity background suppression capability of Lock-in-SIM can be expected to be maintained even under highly challenging experimental conditions.”

27. Qiao, C. et al. Rationalized deep learning super-resolution microscopy for sustained live imaging of rapid subcellular processes. *Nat. Biotechnol.* **41**, 367-377 (2023).

62. Žurauskas, M. et al. IsoSense: frequency enhanced sensorless adaptive optics through structured illumination. *Optica* **6**, 370-379 (2019).

63. Turcotte, R. et al. Dynamic super-resolution structured illumination imaging in the living brain. *Proc. Nat. Acad. Sci. USA*. **116**, 9586-9591 (2019).

64. Liang, Y. et al. Flat-field super-resolution structured illumination microscopy with joint spatial-temporal light modulation. *bioRxiv* (2024).

Comment: Minor Points:

1. In Extended Data Fig. 1, the authors label the bottom of the images “Nucleus.” Are the authors actually sure this is where the nucleus is (i.e. – they have some kind of DNA signal in another channel?). I ask because this is a very unusual ER structure to have under the nucleus, and looks much more like the dense “perinuclear” ER structures that happen around its edges. If they aren’t sure, I would label it more generically as “perinuclear” to avoid an allergic response from someone who studies the nuclear envelope (which does not seem to be visible here).

Response: We thank the reviewer for this reminder. We have corrected the label from “Nucleus” to “Perinuclear” and also revised the figure caption and manuscript accordingly.

Revision:

Extended Data Fig. 1. Reconstruction comparison of the thick cell sample using different 2D-SIM methods. a-d, Experimental ER (labeled with Sec61 β -EGFP) results from widefield microscopy (a), standard 2D-SIM (b), Lock-in-SIM (c), and TIRF-SIM (d). The upper-left diagram shows the corresponding illumination scheme. e,f, Superposition of Lock-in-SIM image (green) with standard 2D-SIM image (magenta) (e) and Lock-in-SIM image (green) with TIRF-SIM image (magenta) (f). The dotted curves denote the **thicker and denser perinuclear** region. Scale bars, 4 μ m (a-f).

(Results, Principle and validation of the background-free Lock-in-SIM, Page 6, Line 174-179)

“To further illustrate the capability and biological benefits of Lock-in-SIM, we focused on the ER in thicker perinuclear regions of the cell (Extended Data Fig. 1) rather than examining only the cell margin. By merging Lock-in-SIM with standard 2D-SIM (Extended Data Fig. 1e) and TIRF-SIM (Extended Data Fig. 1f), Lock-in-SIM clearly preserved the entire ER meshwork while eliminating the background, especially around thicker, more densely packed perinuclear ER.”

Comment: 2. Can the authors make some quantitative estimates from their data about the z-resolution (or effective focal depth?) in their optical sectioning experiments? They demonstrate it is improved compared to 2D SIM, but it would be nice to have a sense of to what degree or at least order of magnitude. The results on the ER in Extended Data Fig. 1 suggest a dramatic improvement in reducing the parts of the organelle sectioned, it would be nice to know how much z-depth that corresponds to.

Response: We thank the reviewer for this constructive suggestion and apologize for the confusion. We would like to clarify that Lock-in-SIM does not improve either lateral (xy) or axial (z) resolution beyond that of standard 2D-SIM; rather, it only enhances the 2D optical sectioning capability by reducing background while preserving the same sample structure. This point has been previously explained in both the Results section “High-fidelity whole-cell optical sectioning reconstruction using Lock-in-SIM with standard 2D-SIM acquisition.” and Methods section “Experimental SIM data acquisition.”

To further substantiate this clarification quantitatively, we specifically acquired both z-sectioning 2D-SIM and 3D-SIM datasets in the revised EdU replication-labelled nuclei puncta experiment. We then analyzed the reconstruction results in both xy and xz views and quantified the z-resolution by calculating the axial full width at half maximum (FWHM) (Supplementary Figs. 17,18). The data indeed corroborated our statements: Lock-in-SIM effectively removes background while maintaining the same structural distribution (Supplementary Fig. 18a), and the z-resolution remains at the diffraction-limited level of approximately 600 nm for both Lock-in-SIM and Wiener-SIM, whereas only 3D-SIM improves it to super-resolved 341 nm (Supplementary Fig. 18b).

These new datasets and analyses have been added to this section and are also provided below for the reviewer’s reference:

Revision:

Supplementary Fig. 17. Optical-sectioning reconstruction comparison of EdU replication-labelled nuclei. **a-c**, Widefield images and corresponding super-resolution images reconstructed using Wiener-SIM and Lock-in-SIM of the same sample at different sample depths. **d-f**, Magnified images of the white boxed regions in **a-c**. From top to bottom, Widefield, Wiener-SIM, and Lock-in-SIM. Scale bars, 3 μm (**a-c**) and 0.5 μm (**d-f**).

Supplementary Fig. 18. 3D reconstruction comparison of EdU replication-labelled nuclei. **a**, Axial views of the widefield images and corresponding super-resolution images reconstructed using Wiener-SIM, Lock-in-SIM, and 3D-SIM of the same EdU sample in Supplementary Fig. 17. **b**, Normalized intensity profiles along the dashed line region in **a**

and the corresponding axial full width at half maximum (FWHM) values. Orange arrows indicate the background artifacts in Wiener-SIM image. Scale bars, 2 μm (a).

Comment: Remarks on code availability:

I have tested both the Fiji plugin and the MATLAB code on approximately 50 images collected by my own group from different SIM implementations. I'd say on 90+% of them, it worked very well. I note it did introduce some artifacts and issues if the input data quality was poor (especially when misalignment of the system caused the underlying SIM illumination pattern to be difficult to make out). All SIM reconstruction patterns have issues with this kind of data, however, so I don't see this as a substantial flaw--but I note it existed and in some cases was exacerbated in the Lock-In SIM compared to simpler Weiner filtered SIM reconstructions.

Response: We would like to express our sincere gratitude to the reviewer once again for recognizing our work and for helping to enhance it through meticulous testing and enlightening and insightful feedback. This comment has been addressed in the first major point above.